# Fast boulder fracturing by thermal fatigue detected on stony asteroids

A. Lucchetti [1] ✉, S. Cambioni[2], R. Nakano[3,34], O. S. Barnouin [4], M. Pajola [1], L. Penasa [1], F. Tusberti [1], K. T. Ramesh [5], E. Dotto [6], C. M. Ernst [4], R. T. Daly [4], E. Mazzotta Epifani[6], M. Hirabayashi [3,7], L. Parro [8,9,10], G. Poggiali [11,12], A. Campo Bagatin [8,13], R.-L. Ballouz [4], N. L. Chabot [4], P. Michel [14,15], N. Murdoch[16], J. B. Vincent [17], Ö. Karatekin[18], A. S. Rivkin [4], J. M. Sunshine [19], T. Kohout [20,21], J.D.P. Deshapriya[6], P.H.A. Hasselmann[6], S. Ieva [6], J. Beccarelli[1], S. L. Ivanovski[22], A. Rossi [23], F. Ferrari [24], C. Rossi[1], S. D. Raducan[25], J. Steckloff[26], S. Schwartz [26], J. R. Brucato [11], M. Dall'Ora[27], A. Zinzi[28], A. F. Cheng [4], M. Amoroso[29], I. Bertini[30], A. Capannolo[16], S. Caporali[11], M. Ceresoli[24], G. Cremonese[1], V. Della Corte[27], I. Gai[31], L. Gomez Casajus [32], E. Gramigna [31], G. Impresario [29], R. Lasagni Manghi [31], M. Lavagna[24], M. Lombardo[31], D. Modenini [31,35], P. Palumbo[33], D. Perna[6], S. Pirrotta [29], P. Tortora [31,35], M. Zannoni [31,35] & G. Zanotti [24]

Spacecraft observations revealed that rocks on carbonaceous asteroids, which constitute the most numerous class by composition, can develop millimeter-to-meter-scale fractures due to thermal stresses. However, signatures of this process on the second-most populous group of asteroids, the S-complex, have been poorly constrained. Here, we report observations of boulders' fractures on Dimorphos, which is the moonlet of the S-complex asteroid (65803) Didymos, the target of NASA's Double Asteroid Redirection Test (DART) planetary defense mission. We show that the size-frequency distribution and orientation of the mapped fractures are consistent with formation through thermal fatigue. The fractures' preferential orientation supports that these have originated in situ on Dimorphos boulders and not on Didymos boulders later transferred to Dimorphos. Based on our model of the fracture propagation, we propose that thermal fatigue on rocks exposed on the surface of S-type asteroids can form shallow, horizontally propagating fractures in much shorter timescales (100 kyr) than in the direction normal to the boulder surface (order of Myrs). The presence of boulder fields affected by thermal fracturing on near-Earth asteroid surfaces may contribute to an enhancement in the ejected mass and momentum from kinetic impactors when deflecting asteroids.

Thermal fatigue is one of the main processes acting on the surface of asteroids causing material damage and failure through the development of fractures[1,2]. Thermal cycling due to changes in insolation generates mechanical stress variations, driving progressive rock breakdown by crack propagation. This process has been widely observed and studied on Earth, Mars, and the Moon[3,4]. The most compelling evidence of thermal fatigue is a preferential north-south (N-S) orientation of boulder-scale fractures, which is in agreement with both models[5] and field observations[6,7].

The effect of thermal fatigue on airless small bodies has been mapped for the first time on near-Earth carbonaceous asteroids (NEA) (101955) Bennu and (162173) Ryugu in high-resolution images acquired by NASA Origins, Spectral Interpretation, Resource Identification and Security-Regolith Explorer (OSIRIS-Rex)[8] and Japan Aerospace Exploration Agency's Hayabusa2 missions[9], respectively. On Bennu, the observation of exfoliation sheets on several boulders[10] and the evidence of preferential meridional orientation of boulders' fractures[11] is consistent with cracking induced by diurnal thermal cycling. On Ryugu, the akin observation of several fractured boulders characterized by an approximately N-S preferred orientation of fractures supports the hypothesis that thermal fracturing plays an important role on asteroid surfaces in general[12]. Nevertheless, while multiple studies reported how thermal fatigue acts on carbonaceous asteroids[10,11,13], our understanding of how thermal fatigue works on S-type asteroids is still limited to laboratory experiments[1,14]. Indeed, the resolution of the imagery data available for Eros and Itokawa is not sufficiently high to map rocks' fractures on S-type asteroid surfaces[15,16].

In this work, we analyze the presence and study the formation of boulders' fractures on Dimorphos, the moonlet of the binary Didymos system, whose surface shows an S-type spectrum[17–19] associated with LL ordinary chondrite material[20]. In addition, dynamical and physical studies suggest that Didymos and Dimorphos should be similar in composition[21,22] since Dimorphos is likely a reaccreted rubble pile formed after mass shedding from Didymos[23]. This asteroid system has been observed over a range of resolutions using the Didymos Reconnaissance and Asteroid Camera for Optical Navigation (DRACO)[24] onboard the DART spacecraft. Here, we make use of the DART Dimorphos images to characterize the fracture length distribution and direction testing the hypothesis that these fractures propagate via thermal fatigue.

The 579-kg DART spacecraft intentionally impacted Dimorphos (177 × 174 × 116 m)[25] on 26 September 2022, demonstrating the technology of kinetic impact redirection[26] and advancing the knowledge of the physical properties of the Didymos binary system[27]. Impact models predict that the ejecta and momentum transfer of kinetic impacts are affected by near-surface porosity[28]. Fracturing via thermal fatigue weakens the surface of a target boulder, which can lower the density of surface rocks with respect to meteorite analogs used in laboratory studies (e.g., LL chondrites[29]). As such, our study of fracture formation on rocks of S-type asteroids is important to improve predictions of the yield of kinetic impactors for planetary defense purposes.

## Results
### Mapping and Analysis of Boulders' fractures
DRACO took many images of the binary asteroid system before impacting Dimorphos. We use the image taken 1.818 s before impact with a pixel scale of 5.5 cm (dart_0401930049_43695_01_iof) covering a portion of the surface of Dimorphos[25]. This is the only image where boulders' fractures are clearly visible and identifiable, along with evidence of boulder morphological variation, "rocks on rocks" and partially buried boulders[24].

In this image, we mapped 54 fractures on 30 boulders (Fig. 1) in an approximately 880 m² area using the Small Body Mapping Tool[30] (SBMT) in order to take into account the shape model[25] and curvature of the asteroid and obtain the correct measured length. To overcome potential observational bias, independent users identified the fractures. Fractures were observed on the boulders with different sizes.

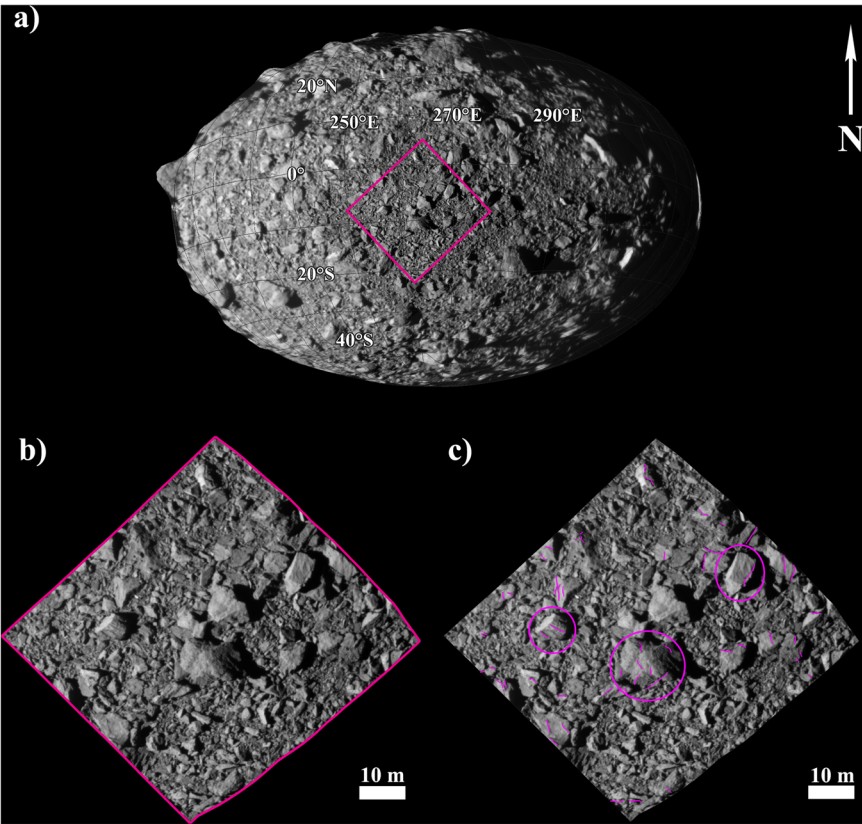

**Fig. 1 | Dimorphos boulders' fractures. a** Dimorphos high-resolution mosaic created with the final 10 DRACO images from 11.447 s to 0.856 s before the impact (pixel scale ranging between 34.9 cm/px and 2.6 cm/px), where the pink box shows the area covered by the image (dart_0401930049_43695_01_iof) analyzed in this work; **b** Close up of the image (dart_0401930049_43695_01_iof) acquired 1.818 s before the impact with a pixel scale of 5.5 cm where boulders' fractures are visible and identifiable; **c** same as b with 54 mapped fractured on different boulders dimension where the modeled boulders are outlined by ellipses. The largest boulder in the scene (6.62 m across), Atabaque Saxum, displays 6 fractures on its surface (Supplementary Fig. 1).

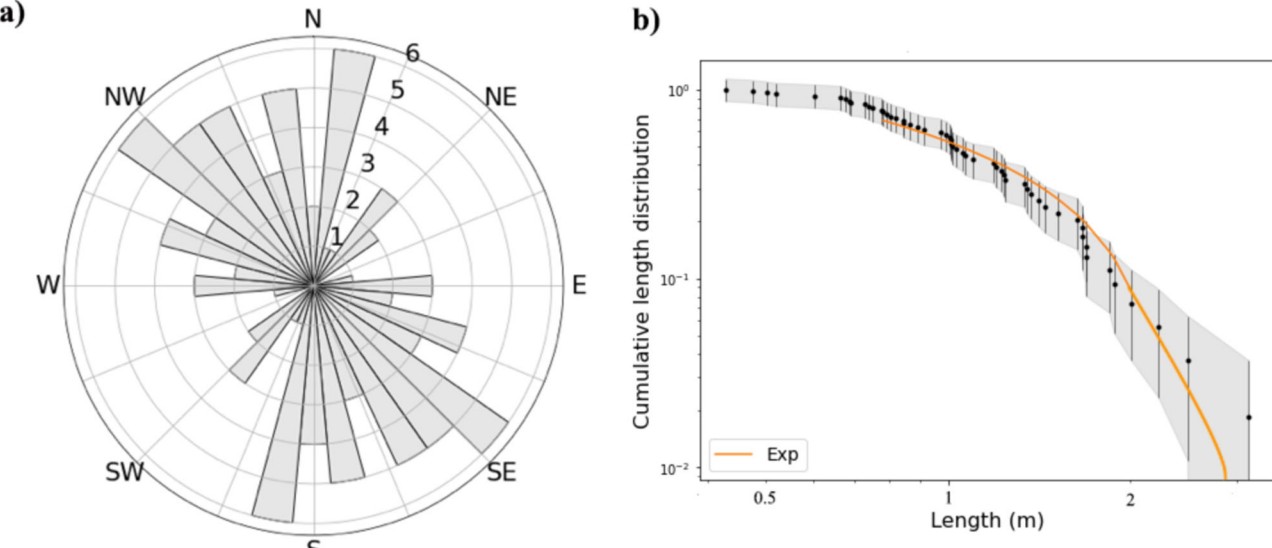

**Fig. 2 | Rose diagram and cumulative distribution of fractures' length. a** Rose diagram showing the NW-SE preferred orientation of Dimorphos boulders' fractures; **b** The fractures' length distribution is well-fitted by an exponential function of the form $N = N_0 e^{bL}$, where $N$ is the fracture cumulative number, and $b$ is a factor that scales the exponent of the function. We derived that the exponential fit is valid for fracture lengths larger than 0.77 m. Source data are provided as a Source Data file.

In the majority of cases, boulders display 1 or 2 fractures. Atabaque Saxum, which is the biggest boulder on the scene (6.62 m), is characterized by 6 fractures on its surface. A large number of fractures are relatively straight with a length ranging between 0.43 and 3.14 m and cross only small portions of the hosting boulders (mean ratio of fracture length to boulder size is 0.4).

The evidence of the thermal-stress formation mechanism comes from the distribution of fractures' orientation with respect to the asteroid north. Assuming that the fractures are formed in situ, they are expected to have a uniform orientation distribution when formed by impacts, while they can be the result of cyclic Sun-induced thermal stresses when presenting a preferred direction orientation, as observed on other Solar System bodies[3,6]. On Bennu, the preferred direction of boulders' fractures has been attributed to mechanical stresses induced by diurnal temperature cycles[10,11]. The preferred orientation in the meridional direction detected on Ryugu's boulders' fractures can also be explained by thermal fatigue[12]. In the same fashion, Fig. 2a displays a non-random orientation distribution for boulder fractures on Dimorphos that are oriented preferentially along a north-west/south-east direction. Dimorphos boulders experience complex thermal insolation due to Sun cyclical exposure that produces diurnal thermal stresses in the east-west direction. This results in the propagation of fractures in the perpendicular direction (north-south). The detection of mapped fractures in Fig. 2a may not be attributed to any mapping detection and observational bias ("Methods" section), hence we can conclude that (i) the fracture formation is due to thermal fatigue and (ii) they have originated in situ on Dimorphos' surface.

Figure 2b shows the length distribution of fractures, which is well-fitted by an exponential law for lengths larger than 0.77 m. To determine the best curve fitting and test its robustness, we followed a statistical approach[31], performing a Kolmogorov–Smirnov test and deriving a p-value equal to 0.6 for the length exponential distribution. This value ensures the validity of our results ("Methods" section). In addition, the fractures are shorter than the hosting boulders, as indicated by the fracture length and hosting boulder size ratio lower than 1 (Supplementary Table 1). Since the resolution of the image analyzed (5.5 cm) is much smaller than the shortest mapped fractures (43 cm), we exclude the presence of any truncation effect affecting the

mapping of short features. An exponential distribution for fractures shorter than the hosting boulders was also previously detected on Bennu's boulder fractures[11] and for polygonal fractures on the nucleus of comet 67P Churyumov–Gerasimenko[32]. In both cases, the exponential distribution was interpreted to be indicative of the role of thermal stress that activates opening-mode fracture propagation within a confined medium without leading to its failure[33].

Based on these findings, we select three different Dimorphos boulders (Supplementary Fig. 1) of different sizes and host a different number of fractures for closer study. We investigate their temperature evolution cycles and apply well-established thermal fatigue models to explore the propagation of fractures. In addition to diurnal thermal stresses, we model the effect of the seasonal temperature variation due to the high heliocentric eccentricity of the Didymos system ($e = 0.38$), resulting in a solar distance variation ranging from -1 to ~2.3 astronomical units (au, "Methods" section). We also account for the effect of eclipses of the primary onto the secondary, which is a unique condition for the binary system, when compared to other small bodies visited by space probes.

## Thermophysical modeling

We model the temperature evolution of the Atabaque Saxum 6.62-m-sized boulder (characterized by 6 fractures), a boulder with 1 fracture (called boulder 1, 5.29 m across), and a boulder with 2 fractures (named boulder 2, 3.51 m in diameter) using a Finite Element Method (FEM) thermophysical model[34]. This model takes a shape model and thermal inertia, $\Gamma$, of a boulder (that is, its resistance to change temperature) as inputs and solves the 3-dimensional heat conduction problem by accounting for scattering sunlight, self-heating due to thermal radiation, shadowings, and the complex mutual dynamics of the binary system. Didymos' thermal inertia was constrained by James Webb Science Telescope (JWST) astronomical observations[35] to be $\Gamma = 260 \pm 30\,\mathrm{J\,s^{-1/2}\,m^{-2}\,K^{-1}}$. However, Dimorphos and Didymos may not have the same $\Gamma$, and $\Gamma$ can change as a function of heliocentric distance[36]. As such, we consider two different values of thermal inertia for Dimorphos: $\Gamma = 370\,\mathrm{J\,s^{-1/2}\,m^{-2}\,K^{-1}}$, which is close to that of Dydimos[35], and $\Gamma = 1000\,\mathrm{J\,s^{-1/2}\,m^{-2}\,K^{-1}}$, which is close to that of the rocky areas of S-type asteroid (25143) Itokawa derived from ground-based infrared observations[37] ($\Gamma_{rock} \sim 900\,\mathrm{J\,s^{-1/2}\,m^{-2}\,K^{-1}}$). We assume that all

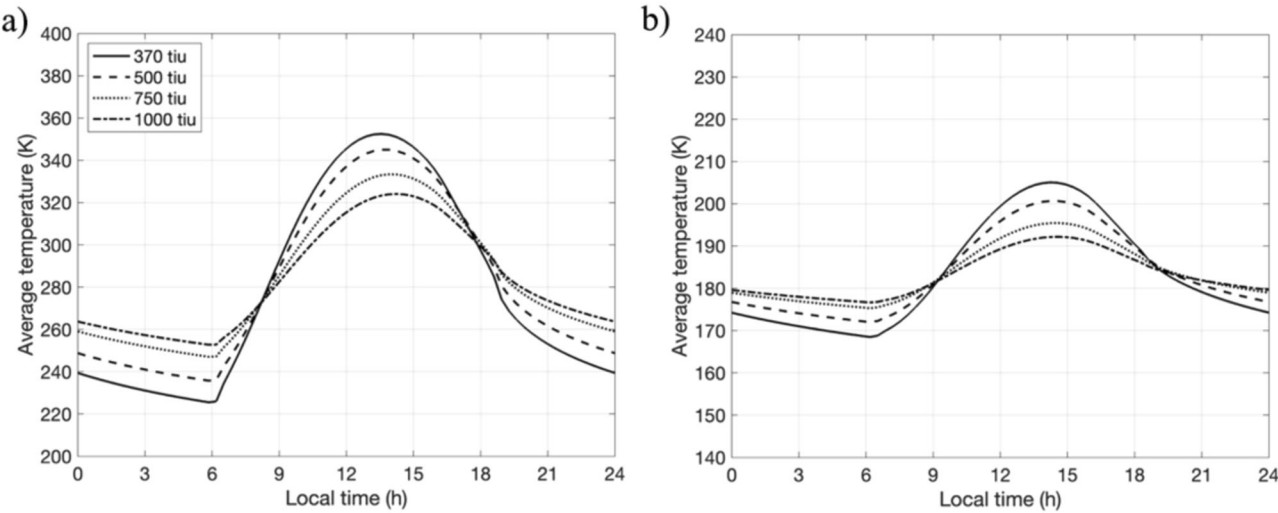

**Fig. 3 | Atabaque's diurnal temperature evolution profile for different thermal inertia values. a** The area impacted by DART on Dimorphos at periapsis. **b** Same area at apoapsis. The temperature at a given time is calculated as the average of the nodal temperatures across Atabaque Saxums's surface.

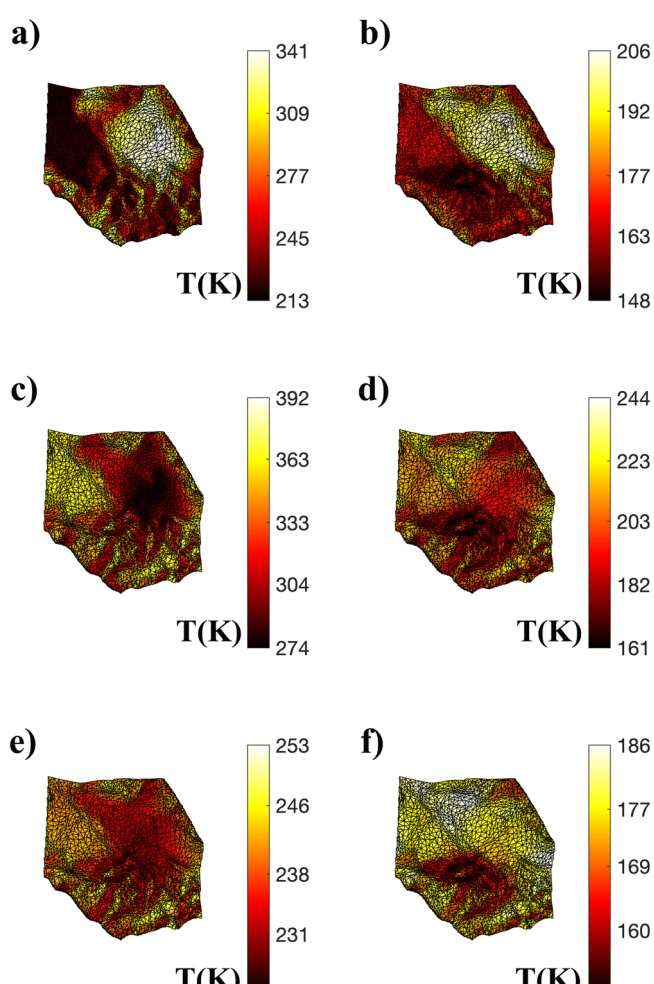

**Fig. 4 | Atabaque Saxum temperature map variation. a**, **c**, and **e** show Atabaque's morning (8 AM), afternoon (4 PM), and midnight (12 AM) temperatures at periapsis, while **b**, **d**, and **f** are the same but for apoapsis.

three boulders on Dimorphos share the same thermophysical properties and that these properties are uniform within each boulder ("Methods" section), neglecting the thermal stress between the elements of the rocks.

The diurnal surface temperature evolution profiles of Atabaque Saxum at the periapsis of Didymos' orbit (Fig. 3a; 1.01 au) show a diurnal temperature variation ($\Delta T$) of ~150 K for $\Gamma = 370 \, \mathrm{J \, s^{-1/2} \, m^{-2} \, K^{-1}}$. At $\Gamma = 1000 \, \mathrm{J \, s^{-1/2} \, m^{-2} \, K^{-1}}$, $\Delta T$ reduces to approximately 80 K, about half of the value at $\Gamma = 370 \, \mathrm{J \, s^{-1/2} \, m^{-2} \, K^{-1}}$. On the other hand, at the apoapsis (Fig. 3b; 2.28 au), Atabaque experiences significantly lower $\Delta T$ values. At $\Gamma = 370 \, \mathrm{J \, s^{-1/2} \, m^{-2} \, K^{-1}}$, $\Delta T$ is approximately 50 K, while at $\Gamma = 1000 \, \mathrm{J \, s^{-1/2} \, m^{-2} \, K^{-1}}$, it drops to about 20 K. The variation of the Atabaque Saxum temperature map is shown in Fig. 4. Owing to the high eccentricity of the Didymos system ($e = 0.38$), Atabaque Saxum experiences approximately 100 K seasonal variations in both $T_{avg}$ and $\Delta T$ for $\Gamma = 370 \, \mathrm{J \, s^{-1/2} \, m^{-2} \, K^{-1}}$ (Fig. 5). This seasonal variation would be smaller for higher $\Gamma$ values. Boulder 1 and boulder 2 exhibit temperature evolution profiles similar to that of Atabaque Saxum due to their close proximity.

Reports suggest that Dimorphos might have been in a tidally locked state prior to the DART impact, based on the assumption that the pre-impact shape is elongated[38], however, DRACO captured its unexpected oblate shape[25]. Given the limited images from DRACO available for detailing Dimorphos's spin state, incorporating orbital data into our thermal model is challenging. Thus, we introduce the assumption that Dimorphos is still tidally locked at the pre-impact phase regardless of its oblate shape. This assumption causes a condition that the asteroid periodically encounters a substantial temperature drop due to the eclipse (e.g., ~100 K drop for $\Gamma = 370 \, \mathrm{J \, s^{-1/2} \, m^{-2} \, K^{-1}}$). However, given the boulder locations at/near the impact site, approximately perpendicular to the Didymos-facing-side, we find that the three boulders remain unaffected by the eclipses, which then are unlikely to have played a role in the propagation of fractures.

### Thermal fatigue modeling

Spacecraft exploration of asteroids[10,11,13], laboratory studies[1], and numerical models[14] have shown that fatigue fractures may develop on asteroid rocks as a response to mechanical stresses generated by temperature cycling. This fracture mechanics is controlled by the thermal inertia, $\Gamma$, the rock size, and the degree of insolation[14].

We input the solution from the thermophysical model into established thermal fatigue models[1,11,13,14] ("Methods" section) to model the propagation of fractures in boulders in the direction towards the

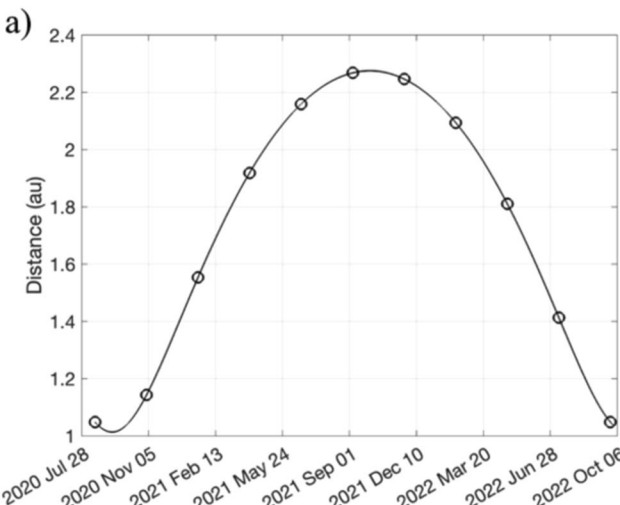

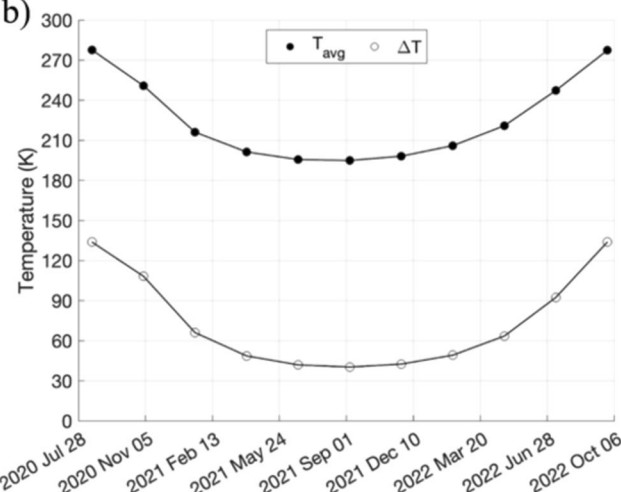

**Fig. 5 | Orbital variation of heliocentric distance and of Atabaque's average diurnal temperature and ΔT. a** The heliocentric orbit period of the Didymos system is 770 days. The open circles indicate the position at which the thermo-physical simulation is conducted. The illumination geometry (i.e., the Sun's center of the asteroid (normal depth direction z) and in the direction of position with respect to the system) was sampled every 70 days from periapsis along the system's heliocentric orbit using a SPICE kernel. **b** The average diurnal temperature and ΔT variations over the heliocentric orbit.

center of the asteroid (normal depth direction z) and in the direction of preferential propagation of fracture (horizontal fractures in direction x). We study the effect of diurnal temperature variation (ΔT) both at aphelion and perihelion to account for seasonal effects but neglect the effect of binary eclipses following the thermophysical modeling results. As such, the estimates of time to break the rock ($t_B$) provided herein are to be considered upper limits. We assume that Dimorphos rocks have properties of LL ordinary chondrites, which is the composition of S-type asteroid Itokawa[39] (Table 1).

We find that on Dimorphos the time of vertical fracture propagation is $t_{B,z}$ ~ 1–10 million years (Myr) for boulders of size equal to 1 m, which is the smallest size of boulders found to host fractures (Fig. 6). This timescale is consistent with extrapolation of $t_B$ for ordinary chondrites[1] and high-thermal-inertia, low-porosity rocks on asteroid Bennu[13]. The value of $t_{B,z}$ becomes comparable to the age of the solar system for a boulder the size of Atabaque Saxum. Most of the vertical propagation of the fracture likely happened during the time of the Didymos system in near-Earth space, where thermal fatigue proceeds ~100× faster than in the main belt[1]. This reasoning suggests that the fractures observed on Dimorphos' boulders are shallow and that the

vertical propagation mode is not the dominant mode of fracture propagation.

Indeed, we find that the propagation of fractures along the boulder surface on Dimorphos may occur one to two orders of magnitude faster than in vertical direction (that is, $t_{B,x}$ ~ 0.01–0.1 $t_{B,z}$; Fig. 7). The cumulative size-frequency distribution of fractures parallel to the surface evolves from that of microcracks observed in meteorites[40] to resemble the size-frequency distribution of the host boulders in ~50–100 kyr assuming thermal inertia of 370 J s$^{-1/2}$ m$^{-2}$ K$^{-1}$, close to the value derived from astronomical observations of Dydimos[35] (~260 J s$^{-1/2}$ m$^{-2}$ K$^{-1}$). This timescale can be as low as 10 kyr if the boulders of Dimorphos have Γ = 1000 J s$^{-1/2}$ m$^{-2}$ K$^{-1}$ (at perihelion), which is similar to the thermal inertia of boulders on S-type asteroid Itokawa derived from ground-based infrared observations[37] (~900 J s$^{-1/2}$ m$^{-2}$ K$^{-1}$). These results are analogous to those from ref. 11. which investigated the formation of fractures via thermal fatigue, but on the carbonaceous asteroid Bennu.

## Discussion

The length distribution of boulders' fractures following an exponential fitting curve coupled with the indication that fractures display a NW-SE direction both support the interpretation that the observed fractures formed through thermal fatigue. This was previously identified on Bennu[11] and Ryugu[12] carbonaceous NEAs, but this is the first time we have this evidence for an S-type binary asteroid. In particular, the preferred orientation of boulders' fractures suggests that such features have originated in situ on Dimorphos, and are inconsistent with the idea that the bulk of the fractures may have formed on Didymos boulders and later transferred to Dimorphos.

Moreover, the fractures' directionality excludes the possibility that most fractures formed as a consequence of impacts, which is also supported by the absence of radial fractures and other signs of impact on the investigated boulders. Indeed, only two craters have been found on Dimorphos boulders[23]. For all such reasons, thermal fatigue is the most feasible process that led to the fracture development we see today on the surface of Dimorphos boulders.

Both diurnal and seasonal temperature cycles contribute to the evolution of the asteroid surface providing smaller or larger temperature variations across the boulders themselves. In particular, thermal stresses generate the propagation of fractures faster in the horizontal direction with respect to the vertical one, constraining

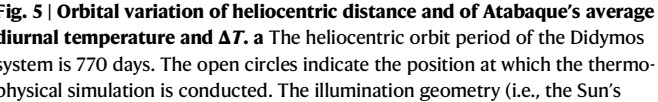

**Table 1 | Material properties assumed for the thermal fatigue model of Dimorphos' boulders**

| Material property | Value (Units) | Reference |
|---|---|---|
| Thermal inertia, Γ | (370, 1000) J s$^{-1/2}$ m$^{-2}$ K$^{-1}$ | Ref. 35. |
| Bulk Density | 3150 kg/m³ | Ref. 25. |
| Heat capacity | 570 J/KgK | Ref. 1. |
| Bond Albedo | 0.067 | Ref. 55. |
| Emissivity | 0.9 | Ref. 36. |
| C (Paris Law prefactor) | 3e−4 | Ref. 1. |
| n (Paris Law exponent) | 3.84 | Ref. 1. |
| Young's modulus | 28.8 GPa | Ref. 56. |
| Poisson ratio | 0.2 | Ref. 11. |
| Thermal expansion coefficient | 8.5e−6 | Ref. 11. |
| Initial fracture size (horizontal fracture) | Lognormal 1e−6–4e−4 | Ref. 40. |

The values are determined from laboratory experiments and analogy with asteroid simulant materials.

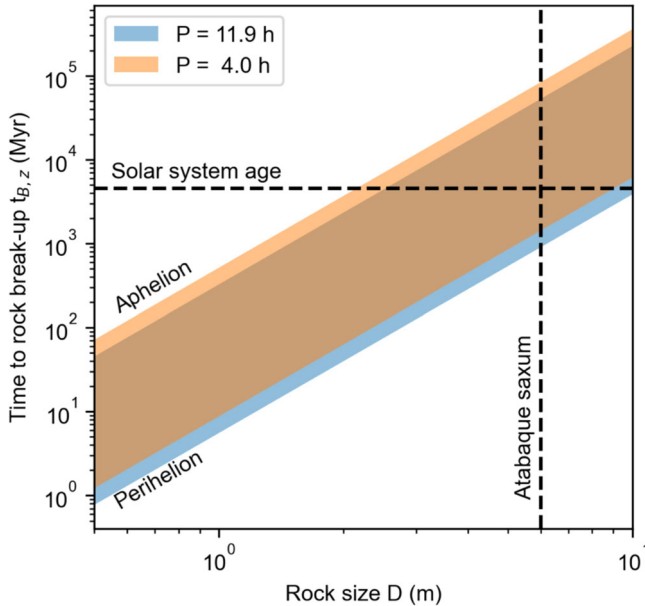

**Fig. 6 | The thermal fractures on Dimorphos' boulders are shallow.** Time to break up $t_{B,z}$ (in units of million years, Myr) as a function of rock size $D$. These results are obtained by scaling the results for ordinary chondrites from ref. 1,11. to the rotation period ($P$) and orbit of the Didymos system ("Methods" section). We adopt the material properties in Table 1.

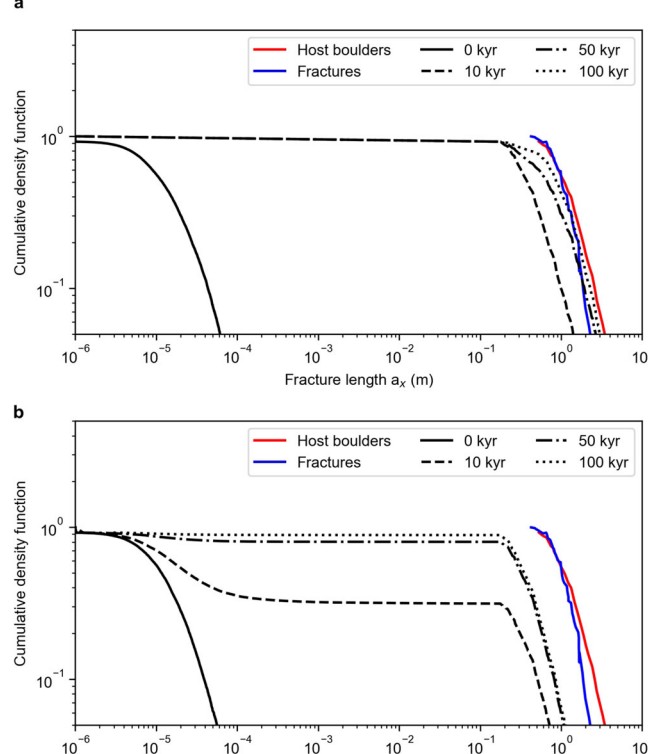

**Fig. 7 | The thermal fractures on Dimorphos' boulders may have developed in less than 100,000 years. a** Cumulative distribution of the length $a_x$ of the horizontally propagating fractures in a boulder of thermal inertia 370 J s$^{-1/2}$ m$^{-2}$ K$^{-1}$ and **b** of thermal inertia 1000 J s$^{-1/2}$ m$^{-2}$ K$^{-1}$. The fractures' initial size-frequency distribution resembles that of microcracks observed in meteorites and evolved to become similar in slope to that of the host boulders and the measured fractures in 10–100 kyr. This time is likely an upper limit because we neglect microscopic stress between elements of the rocks and the effect of eclipses ("Methods" section). The black curves correspond to the average of propagation at aphelion and perihelion with a 3:1 weight. We adopt the material properties in Table 1.

their development on Dimorphos' boulders between 10 and 100 kyr. The time of fracture development should be shorter if the effect of eclipses is included, since such an effect can drop the temperature by 100 K in one hour, hence increasing the stresses on the boulders. However, this does not apply to the area under study because it is perpendicular to the Didymos-facing side and not influenced by such contribution for the current orbital configuration, making our findings an upper limit for fracture development and propagation. We propose that fractures are likely shallow considering the faster rate of horizontal fracture propagation that is the dominant mode of propagation with respect to the vertical one. This is consistent with the relatively young age of Dimorphos that ranges between 0.09 and 0.3 Myr, as inferred from crater size-frequency distribution[23]. In addition, the fracture vertical propagation likely occurred during the time of evolution of the binary system in near-Earth space, where the thermal fracturing rate is two orders of magnitude higher than in the Main Belt[1].

Other mechanisms can be invoked in explaining the formation of boulder fractures, such as impact cratering or material transfer within the binary asteroid system. However, impact-driven mechanisms are inadequate to explain the formation of m-size boulder fractures. Indeed, recent chronology modeling suggests probable impacts less than ~6 cm diameter occurring at least once over 100 kyr if the impact speed is ~15 km/s, under the assumption that the Didymos system has stayed in the near-Earth region over this timescale[38]. Assuming that craters produced by such small projectiles form in the strength regime, the generated crater is likely less than 1 m, according to strength crater scaling rules, which is smaller than the sizes of recorded fractures. Another mechanism can be related to the fact that Didymos is currently rotating at a spin period of 2.26 h, hence, higher centrifugal forces may shed materials from the asteroid and make them reach Dimorphos[41], causing potential resurfacing. Given the recent finding of the Didymos system's bulk density[42] equal to 2.94 g/cm³, which has been further updated from earlier measurements[25], directly shed materials driven by centrifugal effects are unlikely to reach Dimorphos. Unless Didymos spun much faster

over the last 100 kyr, material exchange between Didymos and Dimorphos is also insufficient to explain the fracture formation.

The determination of the thermal fracturing rate on small bodies is important to understand their evolution considering that fatigue depends on boulder properties[10,11,13], rotation rate, and heliocentric distance of the bodies[43]. Indeed, this process could also lead to the disruption of asteroids at small perihelion distance[44], or ejection of particles[45] that could then contribute to the delivery of material to Earth from meteoroid population, making important the exploration of how thermal fatigue acts on small bodies of different composition. The DART mission provided an opportunity to investigate and characterize the Dimorphos boulders' fractures, to find evidence that thermal breakdown contributes to the wide diversity of boulder morphologies and properties on S-type asteroid surfaces, which was observed only on carbonaceous asteroids before. Moreover, exploring the consequence of thermal fatigue has important implications for planetary defense too, as the impact ejecta response from a kinetic impactor, such as the DART spacecraft on Dimorphos[46,47], is influenced by the presence of surface boulder fields. Indeed, the effect of modification of boulder properties due to thermal fatigue adds complexity to the correct prediction of the impact response of boulders-rich surface for future planetary defense missions. Such findings will be tested with the observations coming from the upcoming ESA HERA mission[48] that will provide high-resolution images of the entire surface of both Didymos and Dimorphos.

## Methods

### Mapping, length analysis, and orientation of fractures

We manually identified fractures on Dimorphos boulders on the DRACO image acquired 1.818 s before the impact (dart_0401930049_43695_01_iof). The image has a pixel scale of 5.5 cm and was acquired at a distance of 11.14 km with the solar azimuth near the impact point of approximately −10°N, 60°E. The area covered by the image is approximately 880 m², and it is centered at latitude and longitude of −9.2° and 263.5°E, respectively. Among the different images acquired by DRACO, it is the only one with a resolution that allows the reliable identification of fractures. The fractures were mapped by three different coauthors to ensure the validity of our dataset, finding that 94% of fractures were identified by all mappers. We measured the length of each fracture, along with the dimension of the hosting boulder to calculate the fracture length and hosting boulder size ratio, finding that in each case fractures are smaller than the hosting boulder diameter (Supplementary Table 1).

The fractures' length distribution analysis was investigated by applying the ref. 31. approach to determine their best fitting. We tested both power-law and exponential fit, finding the latter is the most suitable. Indeed, we generated a large number of synthetic datasets and performed 4000 Kolmogorov–Smirnov (KS) tests to verify if generated (power-law and exponential data) and observed data come from the same distribution. This methodology provides a p-value that can be used to quantify the plausibility of the hypothesis. Considering the significance level of 0.10, if the *p*-value is greater than 0.1, we conclude that any difference between the empirical data and the model can be explained with statistical fluctuations, while if the *p*-value is less than 0.1, we conclude that the datasets do not come from a power-law distribution. We obtained a *p*-value for power law equal to 0.1, while a *p*-value for the exponential equal to 0.6. Such results support that the fracture length distribution is well-fitted by an exponential law for a length larger than 0.77 m, supporting the thermal fatigue hypothesis for their formation. In addition, our distribution is not affected by (i) the truncation effect due to image resolution, considering that the shorter fracture mapped is 8 times larger than the image resolution, and (ii) the censoring effect, due to the sampling area compared to the dimension of the longest fracture.

### Lineament orientation biases caused by available lighting

It is well documented that concerns about lighting and viewing conditions can obscure the orientation and number of surface lineaments that are identified on the surface of an asteroid[15]. In order to understand how lighting conditions may play a role in the measurement of fracture orientation presented in this study, we illuminated a local digital terrain model (DTM) of a boulder produced from data collected at Bennu using the OSIRIS-REX Laser Altimeter[49] (OLA). The boulder was chosen because it has numerous linear structures on its surface, in all directions. These structures are well captured by the high-resolution, 5 cm ground sample distance OLA-derived DTM.

For the purpose of this analysis, the local boulder DTM was rotated and translated such that its surface could be illuminated with a range of solar azimuths. The heights of the DTM were aligned along an x-coordinate, while the plane of the DTM was placed along the y-z plane (Supplementary Fig. 2). A sun angle of 0°E, 0°N meant the sun was immediately overhead. The DART solar azimuth near the impact point was approximately −10°N, 60°E. We did not change the camera viewpoint in this assessment, as the emission angle for DART was not very large.

Our analysis reveals that for lighting conditions achieved at the impact point (−10°N, 60°E azimuth), the DART images allow for easy identification of many of the lineaments seen on boulders. Such lighting allows identification of all lineaments that are oriented in N-S, NE-SW, and NW-SE, but could prevent the identification of E-W structures.

Considering the location of our study area (between 5°N and −25°S), the lineaments that should be easily detected are those oriented in N-S, NE-SW, and NW-SE, while the E-W ones can be hidden. In our sample, we detected a small number of lineaments (2 or 3) in both the NE-SW and E-W directions, even if the lighting conditions were favorable to detect NE-SW features. Hence, we consider that the NW-SW fractures preferential direction found may not be an observational bias, since such a result should be valid even if the number of E-W detected lineaments is underestimated.

The rendering of the Bennu boulder reported in Supplementary Fig. 2 indicates to readers that we understand there is an issue with plausible bias, as well as in other studies previously published on lineaments and fractures on asteroids. However, we strengthen the validity of our results with some mitigated confidence with a statistical analysis where we support that our dataset is (i) unlikely to be simply the result of a uniform distribution and (ii) is unlikely to result from a uniform distribution altered by the illumination bias alone (Supplementary Information).

### Thermophysical modeling of binary asteroids

Dimorphos, as it orbits Didymos, encounters complex thermal conditions. It experiences both eclipses (where Didymos casts a shadow over it) and heating from Didymos. Furthermore, the irregular shapes of the bodies and their close proximity induce a strong spin-orbit coupling, referred to as the full two-body problem (F2BP). This dynamic interaction can lead to a complex spin state for Dimorphos.

In our study, we have integrated a thermophysical model[34] and an F2BP dynamics model[50]. The thermophysical model solves the weak form 3-dimensional heat equation via a FEM approach (see below), explicitly accounting for heat conduction between neighboring FEM elements. The F2BP dynamics model also employs an FEM approach to approximate the mutual gravity potential between the bodies[51]. The integration of the two models enables us to simultaneously simulate the thermal condition and the dynamics of Dimorphos. In this section, we briefly introduce key equations for the thermophysical model. For more details on the thermophysical model and the F2BP dynamics model, readers should refer to refs. 34,50, respectively.

Applying Galerkin's method, the integral form of the discretized, weak form of the 3-dimensional heat equation is written as

$$\int_{\Omega^e} \left\{ \phi \rho C_p \frac{\partial T}{\partial t} + \kappa (\nabla T)(\nabla \phi) \right\} dV + \int_{\Psi^e} Q \cdot n\phi \, dS = R, \quad (1)$$

where $\rho$ is the density, $C_p$ is the specific heat capacity, and $\kappa$ is the thermal conductivity, $T$ is the temperature, $V$ is the volume, $S$ is the area, $Q$ is the heat energy at the boundary, and $n$ is the outward positive unit normal vector on the surface. $\phi$ is an arbitrary test function, and $R$ is the residual vector, which is minimized to zero at a solution. $\Omega$ and $\Psi$ denote the volume and surface integrals, respectively. The superscript $e$ indicates the discretized domain; Eq. (1) thus describes the thermal condition of a tetrahedral element $e$.

Using a linear shape function, $N_i^e$, and temperatures of the four nodes composing an element $e$, $T_i^e$, we can approximate the terms inside the integrals in Eq. (1). The element temperature can be approximated as

$$T^e \approx \sum_{i=1}^{4} N_i^e T_i^e = N^e T^e. \quad (2)$$

Using the same shape function, the other terms can be written as

$$\phi^e \approx N^e \phi^e, \tag{3}$$

$$\frac{\partial T^e}{\partial t} \approx N^e \dot{T}^e, \tag{4}$$

$$\nabla T^e \approx \nabla N^e T^e = P^e T^e, \tag{5}$$

$$\nabla \phi^e \approx \nabla N^e T^e = P^e \phi^e, \tag{6}$$

where

$$P^e = \begin{bmatrix} \frac{\partial N_1^e}{\partial x} & \frac{\partial N_2^e}{\partial x} & \frac{\partial N_3^e}{\partial x} & \frac{\partial N_4^e}{\partial x} \\ \frac{\partial N_1^e}{\partial y} & \frac{\partial N_2^e}{\partial y} & \frac{\partial N_3^e}{\partial y} & \frac{\partial N_4^e}{\partial y} \\ \frac{\partial N_1^e}{\partial z} & \frac{\partial N_2^e}{\partial z} & \frac{\partial N_3^e}{\partial z} & \frac{\partial N_4^e}{\partial z} \end{bmatrix}. \tag{7}$$

Using Eqs. (2)–(7), we can rewrite Eq. (1) as

$$A^e \dot{T}^e + B^e T^e + C^e = 0, \tag{8}$$

where

$$A^e = \int_{\Omega^e} \rho C_p N^{e\top} N^e \, dV, \tag{9}$$

$$B^e = \int_{\Omega^e} \kappa P^{e\top} P^e \, dV, \tag{10}$$

$$C^e = \int_{\Psi^e} Q \cdot n N^{e\top} \, dS. \tag{11}$$

Assembling the element matrices and vector (Eqs. (9) – (11)) into the global matrices and vector can be performed in a standard manner described in literature. Once this process is done, we obtain a global system of ordinary differential equations

$$A\dot{T} + BT + C = 0, \tag{12}$$

where $A$, $B$, and $C$ are known as the heat capacity matrix, thermal conductivity matrix, and thermal load vector, respectively.

Equation (12) can be numerically solved by imposing surface and internal boundary conditions. The internal boundary condition is given by

$$(\nabla T)_{l \to \infty} \to 0, \tag{13}$$

where $l$ denotes the depth from the surface. By expanding Eq. (11) for surface elements, the surface boundary condition can be written as

$$C^{se} = \int_{\Psi^{se}} Q \cdot n N^{e\top} \, dS = \int_{\Psi^{se}} f \cdot n N^{e\top} \, dS + \int_{\Psi^{se}} \varepsilon \sigma T^{se4} N^{e\top} \, dS, \tag{14}$$

where $\varepsilon$ is the emissivity, $\sigma$ is the Stefan-Boltzman constant, and $T^{se}$ is the temperature of the surface element. $f \cdot n$ and $\varepsilon \sigma T^{se4}$ describe the solar flux and radiation contributions, respectively. The solar flux term includes four components: direct solar flux $U$, diffuse solar flux $W$, direct self-heating $u$, and diffuse self-heating $w$. The formulations for these terms are detailed in, for example, ref. 52.

We propagate the positions and orientations of Didymos and Dimorphos using the F2BP model and perform ray-triangle intersection tests at each simulation timestep, in order to identify eclipse occurrences and update the surface boundary condition. In this work, we assume that the direct solar radiation scattered off a facet and reaching another facet is entirely absorbed at the receiving facet (i.e., single-scattering mode). Thus, the diffuse self-heating contribution $w$ is set to zero. It is worth noting that the Didymos system is assumed to be in a dynamically relaxed state prior to the DART impact. In this state, Dimorphos is tidally locked with minimal libration[38]. Under such stable conditions, we find the F2BP has a negligible influence on Dimorphos's surface temperature evolution. Additionally, due to the consistent occurrence of eclipses only on the Didymos-facing side of Dimorphos, the three boulders located at or near the DART impact site remain unaffected by the eclipses. Lastly, we neglect the heating from Didymos, as it induces minor changes in surface temperature only on the Didymos-facing side by a few percent (e.g., ~10 K at midnight for periapsis), which is below the temperature difference caused by variations in Γ values[53].

## FEM layer mesh model

To efficiently simulate the thermal condition of a global shape or a local topographic feature under consideration, the thermophysical model employs a unique shape model called FEM layer mesh[34]. The FEM layer mesh consists of thin layers of 4-node tetrahedral volume elements for the surface and subsurface, while the interior is left empty. This approach can be used when the diurnal heat penetration depth is substantially smaller than the radius of the target body or local topographic feature. The thermal conduction at a sufficiently deeper region is, therefore, not affected by the surface thermal condition or vice versa. The diurnal heat penetration depth is the depth at which the temperature variation amplitude is decreased by a factor of $1/e$, given by

$$l_s = \frac{\Gamma}{\rho C_p} \sqrt{\frac{P}{\pi}} \tag{15}$$

where $P$ is the spin period of the body. We typically set the first few layers from the surface to have ~$0.5 \times l_s$ and the total thickness of the FEM layers to be substantially larger than $l_s$, in order to properly resolve the surface and subsurface thermal conditions. The surface nodes are subject to the boundary condition consisting of direct and diffuse solar flux as well as direct self-heating[52], while most inner nodes are constrained to have zero temperature variation.

We construct the FEM layer mesh models of the three boulders from the mission-derived DEM from SBMT (l_00050mm_spc_dtm_dimo_0884s26430_v00325), although the resolution is reduced to approximately 20 cm to save the computation burden in the thermophysical simulation. We assume a smooth surface considering that the shape model does not resolve the ~cm scale roughness of the boulders themselves. The diurnal heat penetration depth of Dimorphos is approximately 3 cm. Given this, each FEM layer mesh model has 10 layers with varying thicknesses from 1 to 20 cm from surface to subsurface. The total FEM layer thickness is ~70 cm, which is greater than $l_s$. We record the nodal temperatures of the FEM layer mesh model once the periodic, steady-state temperature distribution is attained after a few tens of rotations (~15 Julian Days).

## Thermal fatigue modeling

As in ref. 11. we assume that a boulder resting on Dimorphos surface is a cylinder of diameter $D$ and length $2L_x$ along a major axis (x) aligned along the preferential direction of fracture propagation (Fig. 2a). The boulder has an initial single planar surface fracture of length $a_z$ in the direction towards the center of Dimorphos, length $2a_x$ in the direction of preferential fracture propagation, and face normal to x. The fracture vertical and horizontal growth rates are independently computed using the Paris–Erdogan law, as in ref. 11.

$$\frac{da_z}{dN_z} = C[\Delta K_{Iz}(x = 0, y = 0, z = a_x)]^n \tag{16}$$

$$\frac{da_x}{dN_x} = C[\Delta K_{Ix}(x = a_x, y = 0, z = 0)]^n \tag{17}$$

where $N$ is the number of temperature cycles, $C$ and $n$ are the Paris law prefactor and exponent (Table 1) and $\Delta K_I$ is the stress intensity factor excursion.

We solve Eq. (2) to compute the time $t_z$ necessary for rock breakup in the vertical direction ($a_z = D$) using methods developed in previous studies[11,13,14]:

$$t_z = NP \propto \left(\frac{D}{l_s}\right)^{n-1} \zeta(E, \rho, C_P, \alpha)^{1-n} \Delta T^{3/2(n-1)} \tag{18}$$

where $N$ is the number of cycles, $P$ is the rotation period of the asteroid, $\zeta$ is a function of the material properties (including the material's Young elastic modulus $E$ and thermal expansion coefficient $\alpha$, Table 1), $\Delta T$ is the diurnal temperature difference on Dimorphos computed using our thermophysical model and we assume $D \gg l_s$. The value of N for ordinary chondrites of size $l_s = D$ can be read from Fig. 1 of ref. 1. which assumes an asteroid's rotation period $P = 6$ h. We scale these values to a generic $D \gg l_s$ and different rotation periods using Eq. (4) to calculate $t_z$. We explore two rotation periods: 11.9 h (as measured for Dimorphos[25]) and 4 h. The latter represents a generic case of a possible, faster-rotating Dimorphos owing to the Yarkovsky–O'Keefe–Radzievskii–Paddack effect. Furthermore, because Didymos has an eccentric orbit, we model thermal fatigue for $\Delta T$ at perihelion (-1 AU, 20 K) and aphelion (-2.5 AU, 80 K), for a total of 4 cases. We report the results in Fig. 6.

Next, we numerically solve Eq. (3) to compute the time $t_x$ necessary for a fracture in the horizontal direction to propagate from a length $a_{x0}$ to a length $a_{xf} < 2L_x$:

$$t_x = PC^{-1} \int_{a_{x0}}^{a_{xf}} K_{Ix}^{-n} da_x \tag{19}$$

where $K_{Ix}$ is the combination of macroscopic thermal stresses generated by the variations of temperature with depth and microscopic thermal stresses due to thermal expansion differences between different components of the materials. Here we assume that the simplifying case that the latter is zero, as done in previous studies[11]. This yields a lower limit for $K_{Ix}$ (see their Eqs. 4–7) and, in turn, an upper limit for $t_x$. We evaluate $a_x$ as a function of time starting with a log-normal distribution of initial fractures ranging between $10^{-6}$ and $4 \times 10^{-4}$ m similar to the distribution of micro fractures observed in meteorites[40]. The fractures are hosted by boulders whose size is randomly drawn from the distribution of boulders observed on Dimorphos[54]. We adopt a timestep of 10 years and evolve the horizontal fractures with the constraints that $a_x < 2L_x$. As in the vertical fracture propagation model, we explore fracture growth at aphelion and perihelion. In addition, we model the effect of two thermal inertia values of $1000 \, \mathrm{J \, s^{-1/2} \, m^{-2} \, K^{-1}}$ and $370 \, \mathrm{J \, s^{-1/2} \, m^{-2} \, K^{-1}}$ (the latter corresponds to $\Delta T$ - 50 K and -150 K at aphelion and perihelion, respectively). We report the results in Fig. 7.

## Data availability
The DART mission archive at NASA's Planetary Data System contains data from DRACO, as well as associated documentation and advanced products, including the shape models of Didymos and Dimorphos (https://pds-smallbodies.astro.umd.edu/data_sb/missions/dart/index.shtml and https://naif.jpl.nasa.gov/pub/naif/pds/pds4/dart_spice/). The Small Body Mapping Tool developed by Johns Hopkins Applied Physics Laboratory contains the shape models of both asteroids with DRACO images and associated back-planes that resolve the surfaces of the asteroids (https://sbmt.jhuapl.edu/). The Dimorphos fractures'

boulders data generated in this study are provided in the Source Data file. Source data are provided with this paper.

## Code availability
Fracture mapping was performed using the SBMT tool, which is available at https://sbmt.jhuapl.edu/. Thermophysical model is available at https://zenodo.org/record/8103007. The thermal fatigue model is available at with https://doi.org/10.5281/zenodo.6373668.

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

## Acknowledgements

This work was supported by the Italian Space Agency (ASI) within the LICIACube project (ASI-INAF agreement n. 2019-31-HH.0) and HERA project (ASI-INAF agreement n. 2022-8-HH.0). S.C. acknowledges funding from the Crosby Distinguished Postdoctoral Fellowship Program of the Department of Earth, Atmospheric and Planetary Science, Massachusetts Institute of Technology. R.N. acknowledges support from NASA/FINESST, United States (NNH20ZDA001N/80NSSC21K1527). R.N. also would like to thank Joseph Ivarson for his assistance in conducting the thermophysical simulation. This work was supported by the DART mission, NASA Contract 80MSFC20D0004. SDR acknowledges support from the Swiss National Science Foundation (project number 200021\_207359). P.M. acknowledges funding support from the French space agency CNES, ESA, and The University of Tokyo. L.P.'s contribution was supported by the Margarita Salas postdoctoral grant funded by the Spanish Ministry of Universities – NextGenerationEU and CIAPOS/2022/066 postdoctoral grant (European Social Fund). OK acknowledges funding support from the PRODEX program managed by the European Space Agency (ESA) with the help of the Belgian Science Policy Office (BELSPO). N.M. acknowledges funding support from the European Commission's Horizon 2020 research and innovation programme under grant agreement No 870377 (NEO-MAPP project) and support from the Centre National d'Etudes Spatiales (CNES), focussed on the Hera space mission. T.K. is supported by Academy of Finland project 335595 and by institutional support RVO 67985831 of the Institute of Geology of the Czech Academy of Sciences. We gratefully acknowledge Prof. Dr. Wilkerson and Kay Wohlfarth for the constructive comments and suggestions that improved the quality of the paper.

## Author contributions

A.L. led the project, the interpretation of the results, the development of the manuscript; A.L., M.P., and L.P. mapped fractures; L.P. performed the statistical analysis about fracture orientation, F.T. compared the independent fracture mapping; S.C. performed the thermal fatigue analysis and contributed to the interpretation of the results, R.N. performed the thermophysical analysis and contributed to the interpretation of the results; A.L., S.C., and R.N. wrote the original draft of the manuscript; O.S.B. performed the observational bias analysis in lineaments detection and contributed to the interpretation of the results; M.P. and K.T.R. contributed to the interpretation of the results; F.T., E.D., C.M.E., R.T.D., E.M.E., M.H., L.P., G.P., A.C.B., R.L.B., N.L.C., P.M., N.M., J.B.V., O.K., A.S.R., J.M.S., T.K., J.D.P.D., P.H.A.H., S.I., J.B., S.L.I., A.R., F.F., C.R., S.D.R., J.S., S.S., J.R.B., M.D., A.Z., and A.F.C. provided comments that substantially revised the manuscript; N.L.C, A.S.R, and A.F.C. led the DART investigation team; E.D. led the LICIACube team; M.A., I.B., A.C., S.C., M.C., G.C., V.D, I.G., L.G.C., E.G., G.I., R.L.M., M.La., M.Lo., D.M., P.P., D.P., S.P., P.T., M.Z., and G.Z. are part of the LICIACube team and reviewed the manuscript.

## Competing interests

The authors declare no competing interests.

## Additional information

[1]INAF-Astronomical Observatory of Padova, Vic. Osservatorio 5, 35122 Padova, Italy. [2]Department of Earth, Atmospheric and Planetary Sciences, Massachusetts Institute of Technology, Cambridge, MA, USA. [3]Daniel Guggenheim School of Aerospace Engineering, Georgia Institute of Technology, Atlanta, GA 30332, USA. [4]Johns Hopkins University Applied Physics Laboratory, Laurel, MD 20723, USA. [5]Johns Hopkins University, Baltimore, MD, USA. [6]INAF-Osservatorio Astronomico di Roma, Monte Porzio Catone, Roma, Italy. [7]Department of Aerospace Engineering, Auburn University, Auburn, AL 36849, USA. [8]IUFACyT. Universidad de Alicante, Alicante, Spain. [9]University of Arizona, Tucson, AZ, USA. [10]Universidad Complutense, Madrid, Spain. [11]INAF-Osservatorio Astrofisico di Arcetri, Firenze, Italy. [12]LESIA-Observatorie de Paris PSL, Paris, France. [13]Departamento de Física, Ingeniería de Sistemas y Teoría de la Señal, Universidad de Alicante, Alicante, Spain. [14]Université Côte d'Azur, Observatoire de la Côte d'Azur, CNRS, Laboratoire Lagrange, Nice, France. [15]School of Engineering, Department of Systems Innovation, The University of Tokyo, Tokyo, Japan. [16]Institut Supérieur de l'Aéronautique et de l'Espace (ISAE-SUPAERO), Université de Toulouse, Toulouse, France. [17]DLR Institute of Planetary Research, Berlin, Germany. [18]Royal Observatory of Belgium, Uccle, Belgium. [19]Departments of Astronomy and Geology, University of Maryland, College Park, MD, USA. [20]University of Helsinki, Helsinki, Finland. [21]Institute of Geology of the Czech Academy of Sciences, Prague, Czech Republic. [22]INAF-Osservatorio Astronomico di Trieste, Trieste, Italy. [23]IFAC-CNR, Sesto Fiorentino, Firenze, Italy. [24]Dipartimento di Scienze e Tecnologie Aerospaziali, Politecnico di Milano - Bovisa Campus, Milano, Italy. [25]Space Research and Planetary Sciences, Physikalisches Institut, University of Bern, Bern, Switzerland. [26]Planetary Science Institute, Tucson, AZ, USA. [27]INAF-Osservatorio Astronomico di Capodimonte, Napoli, Italy. [28]Space Science Data Center – ASI, Roma, Italy. [29]Agenzia Spaziale Italiana, Roma, Italy. [30]Dipartimento di Scienze & Tecnologie, Università degli Studi di Napoli "Parthenope", Centro Direzionale, Napoli, Italy. [31]Dipartimento di Ingegneria Industriale, Alma Mater Studiorum - Università di Bologna, Forlì, Italy. [32]Centro Interdipartimentale di Ricerca Industriale Aerospaziale, Alma Mater Studiorum, Università di Bologna, Forlì, Italy. [33]INAF-Istituto di Astrofisica e Planetologia Spaziali, Roma, Italy. [34]Present address: Department of Aerospace Engineering, Auburn University, Auburn, AL 36849, USA. [35]Present address: Centro Interdipartimentale di Ricerca Industriale Aerospaziale, Alma Mater Studiorum, Università di Bologna, Forlì, Italy. ✉e-mail: alice.lucchetti@inaf.it

