## [Peer Review File · Nature Communications]

Fast boulder fracturing by thermal fatigue detected on stony asteroidsREVIEWER COMMENTS

Reviewer #1 (Remarks to the Author):

I recommend minor revisions of "Fast boulder fracturing by thermal fatigue detected on stony asteroids" by Lucchetti et al. prior to publication in Nature Communications. The manuscript adds to the growing body of literature finding that thermal fatigue is a key process on the surfaces of asteroids, moons, and other airless bodies. The manuscript seems to be inspired by Ref. 11, which reported many similar observations, analysis, and figures for fracture of boulders on Bennu. In the present manuscript, images of Dimorphos' surface obtained from the recent DART mission reveal a Northwest-Southeast preferred orientation of boulder fractures. This NW-SE orientation is fairly consistent with thermoelastic model calculations of diurnal temperature cycling, which predict the tensile normal stresses to be greatest along the East-West direction. The key implication is that these fracture events happened due to thermal weathering, and that these fractures occurred while the boulders resided on the surface of Dimorphos. Overall, the manuscript is adequately well-written with the implications and conclusions being sufficiently well supported by the data and analysis.

A few minor revisions of note.

1. In table 1, there are no units provided for initial crack sizes. Also, the letter zeta is missing missing on line 476. There may be other such typos.
2. Some of the figures seem to be low resolution and some in-line equations are poorly formatted.
3. Please rephrase the statement "We solve equation (2) to compute the time ... ". This gives the impression that Eqn (4) contains something novel by the authors. As far as I can tell, this is exactly the derived solution by Delbo and I from Ref. 11 with no modifications from the current authors. Would it not be less confusing to the reader to simply state that the present predictions are obtained from inputting appropriate material/orbital/model parameters for Dimorphos into the two thermal fatigue models of a_x and a_z that are described in detail in Ref. 11?

Respectfully,

Justin W. Wilkerson, Ph.D.
Associate Professor & Sallie and Don Davis '61 Career Development Professor
J. Mike Walker '66 Department of Mechanical Engineering
Texas A&M University

Reviewer #2 (Remarks to the Author):

Please see the attachment.

Reviewer #2 (Attachment):

The authors analyse the orientation and size distribution of boulder fractures on Dimorphos, an S-type asteroid and the moonlet of Didymos. The data is derived from imagery of the DRACO camera onboard the DART spacecraft, acquired seconds before the spacecraft hit the body. The orientation statistics of the boulder fractures indicate a preferred north-south direction, and the size statistics are best fitted with an exponential function. Considering various forms of observational bias, the authors conclude that both observations are consistent with rock fractures due to thermal fatigue through thermal cycling. The authors run thermal models and rock fatigue models to support this hypothesis.

This analysis is the first of its kind and addresses an interesting topic relevant to the field. The literature review considers all relevant resources. The manuscript merits publication in Nature Communications after the authors addressed the following points.

Major points:

The study must show compelling evidence of a preferred north-south orientation of rock fractures to merit publication. Therefore, I recommend revising the study's design by (1) explicitly stating a null hypothesis, (2) quantifying the effect of observational bias through lightning conditions, and (3) discussing sampling bias. I understand that designing a full significance test within a statistical framework is extremely difficult in this particular science case. However, some numerical estimate of how strong the effect is compared to any bias will make the study more believable and thus much better.

(1) Checking whether some effect holds requires some form of baseline, i.e., a null hypothesis. The authors do not explicitly formulate a null hypothesis but mention that fractures are not formed through thermal fatigue, but impacts are likely uniformly oriented (line 125). Later, the authors use a boulder of Bennu with lineages in all directions to test the influence of lighting conditions. In a sense, the statement in line 125 appears as a null hypothesis, but it already contains an interpretation, namely that impacts played a role. A clean null hypothesis for this study would be simply assuming randomly oriented fractures. The authors should thus explicitly state a null hypothesis early on in their work, such as: ... "As a null hypothesis, we assume that the orientation of rock fractures is uniformly distributed"... and contrast it with the hypothesis in question.

(2) The authors discuss observational bias introduced into the study, such as individual labelling style and illumination conditions. The study on labelling performance is fine. However, my largest concern is about the lightning. Using a boulder of Bennu is a great starting point, but the study and conclusions described in the paragraph starting at line 372 remain too qualitative. I recommend extending this study to obtain a distribution (rose diagram) of how well randomly oriented boulder fractures can be observed under Dimorphos-like lighting conditions as captured by DRACO. The resulting distribution can then be (numerically) compared with the observed distribution (rose diagram, Figure 2). If there is a significant difference, the preferred direction of fractures found in DRACO's imagery is believable.

For extending the investigation of observational bias through different lighting, please use more existing boulders or simulated synthetic boulders with randomly oriented fractures and illuminate them under different angles. You may also consider different boulder lengths, fracture shapes, depths, and widths. Please continue with a perceptual study with multiple experts or a simple automatic detector that senses fractures above a certain threshold. This study should yield a distribution (rose diagram) of how well randomly oriented boulder fractures can be observed under Dimorphos-like lighting conditions. Are there similar studies in the literature? Maybe from different bodies (Moon, Mercury) or even different disciplines like earth science, computer vision, computer visualization, cognition science?

(3) What is missing is whether the preferred N-S direction of boulder fractures results from sampling bias from an otherwise randomly oriented boulder fracturing process. The DRACO camera provided one image with only a small set of boulders. Please investigate whether the shape and alignment of boulders may cause any bias in the direction of boulder fractures. Is there any bias in the given DRACO image? I think this can be achieved with simple simulations.

Minor comments line by line:

Line 132: The formulation “nonlinear temperature gradient” might be ambiguous because temperature gradients can occur in different forms at airless planetary bodies, such as vertical thermal gradients, lateral thermal gradients, and thermal gradients in the first few millimetres of the uppermost regolith. Please specify the type of gradient.

Line 160: Figure 1. I think it would be beneficial to indicate the locations of the boulders described in the text.

Line 178. Please also check if Boulder 1 and Boulder 2 carry a name. The USGS already lists several boulder names, such as Bodhran Saxum, Caccavella Saxum, and Puniu Saxum. <https://astrogeology.usgs.gov/news/nomenclature/five-names-approved-for-dimorphos>

Line 180: Please include a reference to the shape model.

Line 187: The thermal inertia values of $370 \text{ Js}^{-1/2}\text{m}^{-2}\text{K}^{-1}$ and $1000 \text{ Js}^{-1/2}\text{m}^{-2}\text{K}^{-1}$ are reasonable

Line 216: Figure 3 provides the average temperatures of Atabaque Saxum. However, different orientations of the boulder’s surface may lead to different local temperatures. I think it would be very instructive to plot the shape model of the boulder overlaid with a temperature map that highlights the complex spatial distributions of the temperature. The best would be a short sequence with three images: morning, midday, and afternoon.

Line 246 - 248: What does “This” mean? Please specify: This argument..., this reasoning...

Line 250: I think it should be $t_{B,x} \sim 0.01-0.1t_{B,z}$ instead of $t_{B,x} \sim 0.01-0.1t_{B,x}$ because you compare the lateral and the vertical fracture propagation time.

Line 262: Please explain Figure Caption 5 of what P is.

Line 254: “This timescale can be as low as 10 kyr...” I understand that Figure 6 a) presents the cumulative density function of the fracture length for $370 \text{ Js}^{-1/2}\text{m}^{-2}\text{K}^{-1}$, and the model values for 50 kyr and 100 kyr well align with the data. However, I struggle to follow your argumentation for Figure 6 b with $1000 \text{ Js}^{-1/2}\text{m}^{-2}\text{K}^{-1}$. Where does your conclusion “This timescale can be as low as 10 kyr” come from, and where do I find this in Figure 6 b? Please elaborate.

Line 263: Please explain what scaling means in this context.

Line 280: “... because we neglect thermal stress between the elements of the rocks.” Please mention this aspect earlier.

Line 284-341: Please revise the discussion according to the new study design.

Line 414: Which kind of potential? Please specify.

Line 427: Please provide a numerical value that

Line 444: The authors mention that their thermal model uses direct and diffuse self-heating. Self-heating is a comparatively complex problem requiring more details: What is the bolometric albedo for

self-heating? This quantity might be different from the bolometric albedo that determines the solar flux the surface absorbs. Self-heating is used to compute the temperature map T of the surface but already requires the temperature distribution. Consequently, it is a coupled problem. Do you use a full numeric solution to approach self-heating, or do they use a single-scattering approximation?

Line 463: I think the full name is the Paris-Erdogan law.

Reviewer #3 (Remarks to the Author):

This manuscript presents a work regarding new findings on the asteroid Dimorphos that could have important implications on the activity and long term evolution of this type of asteroid, especially in a binary system from which no in-situ mission data existed.

Although the thermal fracturing process underlined by authors has been already recently described on other near Earth asteroids, their findings apply on S-type asteroid for which no fractures observation existed before. Moreover, authors' study seems to confirm that this process is indeed pervasive and of universal relevance, at least for all rubble pile structured asteroids, and it is especially striking in the case of Dimorphos, being of a very small size.

The manuscript is clear and to the point, with relevant figures supporting the claims.

Interpretations are convincing and their implications are adequately discussed. However, considering the data shown and their limitations, I think that authors should moderately tone down their claims on the meridian direction of the observed fractures. Though, I consider that it won't change the overall meaning and relevance of this study.

For the above reasons, I think this work will be worth to be published in Nature Communication and recommend these minor revisions to be addressed before final acceptance.

(for specific comments see in the .pdf attached)

General comments:

This manuscript presents a work regarding new findings on the asteroid Dimorphos that could have important implications on the activity and long term evolution of this type of asteroid, especially in a binary system from which no in-situ mission data existed.

Although the thermal fracturing process underlined by authors has been already recently described on other near Earth asteroids, their findings apply on S-type asteroid for which no fractures observation existed before. Moreover, authors' study seems to confirm that this process is indeed pervasive and of universal relevance, at least for all rubble pile structured asteroids, and it is especially striking in the case of Dimorphos, being of a very small size.

The manuscript is clear and to the point, with relevant figures supporting the claims. Interpretations are convincing and their implications are adequately discussed. However, considering the data shown and their limitations, I think that authors should moderately tone down their claims on the meridian direction of the observed fractures. Though, I consider that it won't change the overall meaning and relevance of this study.

For the above reasons, I think this work will be worth to be published in Nature Communication and recommend these minor revisions to be addressed before final acceptance.

Detailed comments and minor comments:

- Throughout the manuscript the authors use the word "meridional" to qualify the "N-S/along meridian" directions of the thermal fractures. If I'm not mistaken the valid word should be "Meridian" as "meridional" meaning is "in the southern part".

- In Line 121, authors mention the fracture length to boulder size ratio, which is later used to show that all fractures don't cross through whole boulders. Unless I missed it, I could not find in the main text nor methods parts how this boulder size is defined? Is it the longest dimension of the boulder on the image? Or the major axis length of an ellipse fitted to each boulder? Authors should clarify this point. Thank you.

- In Line 233 and in Figure 6 caption, unless I didn't understand well, authors mention "horizontal fractures". As stated by authors in other part of the manuscript and as visible by interpreting the fractures trace onto the boulder geometry, most-to-all of the observed fractures seems to be vertical, but propagating preferentially in the horizontal direction.

Please clarify this point and modify these sentences accordingly.

- At line 146 authors validly mention that thermal polygonal fractures on 67P have been shown to follow "exponential size distribution" and quote the reference 32 following this claim. To my knowledge, this article does not mention the size distribution of 67P polygonal fractures. However, such exponential distribution of polygonal fractures on 67P has been underlined in the following work. <https://doi.org/10.1038/s41561-019-0307-9>

- Throughout the manuscript, the authors state that their analysis clearly shows that fractures are oriented following a "meridian" direction. I think their interpretations is likely valid and well discussed in the method and Sup Material parts. However, considering the small number of interpreted fractures and the possible observation bias that authors validly explain for E-W fracture, I feel that authors need to be a bit more cautious and could use more conditional in the main text too. For instance, by stating

that fractures directions are surely “Non-random”, which won’t change the meaning and relevance of the results. Indeed, I think the non-randomness of fracture direction is the most important point here. Moreover, it could be interesting to discuss the fact that E-W trending fractures could indeed exist, without necessarily impeding the thermal origin of the observed fractures, since other directions could result from previous shifts in the spin axis of Dimorphos. Speaking of which, it may be interesting to discuss possible shift of the spin axis in the past? And is it expected to be rather unstable, considering the binary nature of this system?

REVIEWER COMMENTS

Reviewer #1 (Remarks to the Author):

I recommend minor revisions of "Fast boulder fracturing by thermal fatigue detected on stony asteroids" by Lucchetti et al. prior to publication in Nature Communications. The manuscript adds to the growing body of literature finding that thermal fatigue is a key process on the surfaces of asteroids, moons, and other airless bodies. The manuscript seems to be inspired by Ref. 11, which reported many similar observations, analysis, and figures for fracture of boulders on Bennu. In the present manuscript, images of Dimorphos' surface obtained from the recent DART mission reveal a Northwest-Southeast preferred orientation of boulder fractures. This NW-SE orientation is fairly consistent with thermoelastic model calculations of diurnal temperature cycling, which predict the tensile normal stresses to be greatest along the East-West direction. The key implication is that these fracture events happened due to thermal weathering, and that these fractures occurred while the boulders resided on the surface of Dimorphos. Overall, the manuscript is adequately well written with the implications and conclusions being sufficiently well supported by the data and analysis.

We thank Prof. Dr. Wilkerson Referee for the positive feedback and for useful suggestions. We addressed all the requested revisions below.

A few minor revisions of note.

1. In table 1, there are no units provided for initial crack sizes. Also, the letter zeta is missing on line 476. There may be other such typos.

We reported in the first row the unit for both boulder size and fracture length (meters) and added it in the caption. We checked the entire manuscript to correct typos.

2. Some of the figures seem to be low resolution and some in-line equations are poorly formatted.

We checked the figures resolution (substituting the Fig.1) and corrected the in-line equations.

3. Please rephrase the statement "We solve equation (2) to compute the time ... ". This gives the impression that Eqn (4) contains something novel by the authors. As far as I can tell, this is exactly the derived solution by Delbo and I from Ref. 11 with no modifications from the current authors. Would it not be less confusing to the reader to simply state that the present predictions are obtained from inputting appropriate material/orbital/model parameters for Dimorphos into the two thermal fatigue models of a_x and a_z that are described in detail in Ref. 11?

The Referee is right. We added "as in Ref. 11" in line 540 to avoid any misunderstanding. We kept the equations and not simply referred to Ref.11 because the editor asked to retain equations and methodological details so the works remains self-sufficient.

Respectfully,

Justin W. Wilkerson, Ph.D.

Associate Professor & Sallie and Don Davis '61 Career Development Professor

J. Mike Walker '66 Department of Mechanical Engineering

Texas A&M University

Reviewer #2 (Remarks to the Author):

Please see the attachment.

The authors analyse the orientation and size distribution of boulder fractures on Dimorphos, an S-type asteroid and the moonlet of Didymos. The data is derived from imagery of the DRACO camera onboard the DART spacecraft, acquired seconds before the spacecraft hit the body. The orientation statistics of the boulder fractures indicate a preferred north-south direction, and the size statistics are best fitted with an exponential function. Considering various forms of observational bias, the authors conclude that both observations are consistent with rock fractures due to thermal fatigue through thermal cycling. The authors run thermal models and rock fatigue models to support this hypothesis. This analysis is the first of its kind and addresses an interesting topic relevant to the field. The literature review considers all relevant resources. The manuscript merits publication in Nature Communications after the authors addressed the following points.

We thank the Referee for constructive comments and suggestions that helped improving the overall quality of the manuscript. We reported below the answers to the Referee's major and minor comments.

Major points:

The study must show compelling evidence of a preferred north-south orientation of rock fractures to merit publication. Therefore, I recommend revising the study's design by (1) explicitly stating a null hypothesis, (2) quantifying the effect of observational bias through lightning conditions, and (3) discussing sampling bias. I understand that designing a full significance test within a statistical framework is extremely difficult in this particular science case. However, some numerical estimate of how strong the effect is compared to any bias will make the study more believable and thus much better.

We thank the Referee for the constructive comments and suggestions. As outlined by the Referee, a full significance test within a statistical framework is extremely difficult for this scientific case, nevertheless, we report below an analysis that strengthen the validity of our results and addressed the comments raised in (1) and (2) after point (2).

(1) Checking whether some effect holds requires some form of baseline, i.e., a null hypothesis. The authors do not explicitly formulate a null hypothesis but mention that fractures are not formed through thermal fatigue, but impacts are likely uniformly oriented (line 125). Later, the authors use a boulder of Bennu with lineages in all directions to test the influence of lighting conditions. In a sense, the statement in line 125 appears as a null hypothesis, but it already contains an interpretation, namely that impacts played a role. A clean null hypothesis for this study would be simply assuming randomly oriented fractures. The authors should thus explicitly state a null hypothesis early on in their work, such as: ... "As a null hypothesis, we assume that the orientation of rock fractures is uniformly distributed"... and contrast it with the hypothesis in question.

(2) The authors discuss observational bias introduced into the study, such as individual labelling style and illumination conditions. The study on labelling performance is fine. However, my largest concern is about the lightning. Using a boulder of Bennu is a great starting point, but the study and conclusions described in the paragraph starting at line 372 remain too qualitative. I recommend extending this study to obtain a distribution (rose diagram) of how well randomly oriented boulder fractures can be observed under Dimorphos-like lighting conditions as captured by DRACO. The resulting distribution can then be (numerically) compared with the observed distribution (rose

diagram, Figure 2). If there is a significant difference, the preferred direction of fractures found in DRACO's imagery is believable.

For extending the investigation of observational bias through different lighting, please use more existing boulders or simulated synthetic boulders with randomly oriented fractures and illuminate them under different angles. You may also consider different boulder lengths, fracture shapes, depths, and widths. Please continue with a perceptual study with multiple experts or a simple automatic detector that senses fractures above a certain threshold. This study should yield a distribution (rose diagram) of how well randomly oriented boulder fractures can be observed under Dimorphos-like lighting conditions. Are there similar studies in the literature? Maybe from different bodies (Moon, Mercury) or even different disciplines like earth science, computer vision, computer visualization, cognition science?

We here report the analyses we have accomplished to strengthen the validity of our findings (preferred orientation of boulder's fractures). In particular, we investigate:

- 1) the probability that the observed azimuthal distribution arose by chance from a uniform distribution, i.e. from randomly oriented fractures;**
- 2) the sampling bias effect, implying that fractures with azimuth similar to the sunlight direction are less likely to be identified, and thus are not present from the dataset.**

The observed azimuthal distribution coming from the fractures analysis performed in this work, coupled with a histogram representation is reported in Figure 1. The resulting azimuth mean value and standard deviation, together with and their relative bootstrapped CI (95% confidence interval) are presented:

- Mean: 150.04° , 95% CI: -15.89, 16.75**
- Std: 41.56, 95% CI: -6.92, 12.79**

Figure 1. Left: Rose diagram of fractures analyzed in this work. Right: histogram representation of such fractures.

We investigate how likely it is to generate by chance such a distribution as function of angular values with statistical properties (mean and std) similar to our dataset.

A clean null hypothesis for this study would result in a purely random orientation of fractures (with a process that could produce such a uniform distribution). If this hypothesis is verified, then any clustering in our data would arise by chance. We can establish a null hypothesis when the observed distribution arises from the sampling of a uniform distribution. To reject the null hypothesis, we need to demonstrate that our distribution, or one that is very similar to ours,

cannot be generated by chance. Hence, we sample multiple times a random uniform distribution with the same sample size of the collected data.

To generate a uniform distribution, we first produce an example with the same number of samples as in our dataset, using a uniform distribution from 0° to 180° . The first plot of Figure 2 shows our observed distribution as a reference. Then, we can repeat this sampling for a large number of cases (100000) – some of which are shown below (Figure 2) -- and demonstrate that the likelihood to produce a standard deviation similar to the one we derived is small, i.e. near 0.129 %. This is of course limited from a statistical point of view, but it provides the indication that is quite unlikely that our dataset arises by chance from a uniform distribution (this can be defined with > 99 % confidence).

Figure 2: Original dataset (in red) as a reference and some examples of generated uniform distributions.

Afterwards, we can make a simplified model of the expected effect of the lighting bias. The idea is that for fractures with the same azimuth of the Sunlight direction, the detection is impossible or at least very difficult.

To model this condition, we can start with a uniform distribution, where fractures with an azimuthal orientation similar to the one of the incoming light are more likely to remain undetectable, hence are not inside the final dataset. We report below an example that should clarify the issue, even if we are aware that such analysis is quite limited from a statistical perspective, but still quite illustrative.

The implementation of the model looks like as follows:

```
def random_dataset(N, loc=90, std=10, corrective_factor=1):
    nprob = norm(loc, std)
    numbers = []
    while len(numbers) != N:
        value = np.random.uniform(0,180, 1)[0]
        p = nprob.pdf(value)/nprob.pdf(loc) * corrective_factor
        reject = np.random.choice([True, False], p=[p, 1-p])
        if not reject:
            numbers.append(value)
    return np.array(numbers)
```

The sun illuminates the scene with an azimuth angle that we assume equal to 100° . In particular, after considering that the subsolar point is located -10°N , 60°E with respect to the location of the boulders, the sun azimuth value is 101.51° . Below, we report some plots to have a clear idea of the behavior of the abovementioned simplified model (Figure 3).

Figure 3: Output of the simplified model including the expected effect of the lighting bias.

Afterwards, we run again the simulation we did above by repeating the sampling of such distribution, with the same number of samples of the original dataset. As standard deviation (std) of the gaussian we use a value of 22. As shown in the next plot (Figure 4), it is clear that this std value generates a synthetic dataset with the same standard deviation as the one observed in real data. We highlight that change of such a value will end up with standard deviation values that are less realistic, hence making difficult to reproduce a dataset with statistical properties similar to our original one.

Figure 4: Synthetic dataset with the same std as the one observed in original data.

Such findings suggest that a random process, as the one modelled above, can generate a dataset with a similar std as the original one. Nevertheless, the process has the effect of removing samples around the 100° azimuth, clustering the values in the opposite direction (10°) (Figure 5a). Hence, in terms of average azimuth value, the generated datasets are quite different. In order to help the reader, we rearrange the dataset to be wrapped around a 90° angle, rather than 180° , hence making the dataset more obviously a gaussian (Figure 5b).

Figure 5a and 5b: Synthetic dataset including the illumination bias effect.

This plot demonstrates that it is quite unlikely to generate the observed average orientation of fractures by a uniform process including the illumination bias effect. Specifically, 0.05 times out of 100, we obtain a mean similar to the one of our original dataset. The plot highlights that it is quite difficult to get a mean of -30 from a randomized sample plus the illumination bias. Furthermore, it shows that the illumination bias would create a decrease of observations at azimuth corresponding to the sun azimuth, hence altering the dataset, and moving the average value to the azimuth located at sun azimuth $+ 90^\circ$.

The above analyses support the fact that i) the dataset is unlikely to be simply the result of a uniform distribution, hence leading to the rejection of the null hypothesis (the dataset comes from a uniform distribution), and ii) the dataset is unlikely to result from a uniform distribution altered by the illumination bias alone. Indeed, the direction where the azimuth values are

clustered differs from the one that would be expected with an illumination bias with azimuth $\sim 100^\circ$.

We highlight that from a statistically perspective, such a result does not prove that the dataset cannot be derived from the superimposition of a uniform distribution process and additional effects, rather that our original dataset does not directly come from a uniform distribution, neither from a uniform distribution plus the illumination bias.

Other biases (e.g., boulders' shape) should also probably be considered, but the limited dataset and the lack of new data (until HERA mission, whose arrival will be in December 2026) make a more advanced analysis quite difficult and limited.

We modified the image showing the rendering of Bennu's boulder (Figure 6), as shown below, to provide a clearer representation of the illumination bias effect.

In addition, three independent users mapped the fractures on the simulated image of Bennu's boulder with the same illumination condition of DRACO image deriving the corresponding rose diagrams (Figure 7). Afterwards, the rose diagrams of the three different mapping cases have been compared to the observed Dimorphos distribution finding a significant difference between the results. This support the interpretation that we can have confidence in the preferred direction of fractures found in DRACO's imagery.

Figure 6: The different illumination conditions used to understand how different lighting may play a role when identifying crack orientations. We decided to choose a boulder of Bennu because this one is affected by lineaments oriented in all directions. Moreover, its digital terrain model (DTM) has been produced by the OSIRIS-REX Laser Altimeter (Daly et al., 2017) with a 5 cm ground sampling distance.

Figure 7: Fractures mapping and corresponding rose diagram of Bennu's boulder. The image used for this analysis is the one with akin illumination condition of DRACO images. The mapping has been performed by three different researchers and the resulting rose diagram differs from the one derived from Dimorphos fractures (in red as reference).

The request to do a detailed study on the effects of lighting bias on the identification of lineaments on boulders is definitely a worthwhile study. We feel that our rendering shown of the Bennu boulder indicates to readers that we understand there is an issue with plausible bias, but our statistical analysis discusses the bias effect and mitigate them with some confidence, as stated in this response.

We have added some words stating that we are aware that lighting bias may remain here (as well as in other studies). The methods section has been modified (lines 379 - 409), while we prefer to report the entire statistical analysis in the Supplementary Information, if the Referee agrees.

(3) What is missing is whether the preferred N-S direction of boulder fractures results from sampling bias from an otherwise randomly oriented boulder fracturing process. The DRACO camera provided one image with only a small set of boulders. Please investigate whether the shape and alignment of boulders may cause any bias in the direction of boulder fractures. Is there any bias in the given DRACO image? I think this can be achieved with simple simulations.

We thank the Referee for the interesting comment. There are currently other two papers under revision investigating the shape of boulders. In particular, Colas et al., paper (*Nature Communication, under revision*) conducted an in-depth examination of the last complete image captured by DRACO before the DART impact, to retrieve the boulder dimensions and several dimensionless morphological parameters as the axial ratio (width-to-length ratio). The analysis of 267 selected boulders provide an axial ratio of 0.6 and for 30 well-resolved boulders a value of 0.67. The axial ratio value of all boulders located on the entire Dimorphos has been calculated in the Pajola et al., paper (*Nature Communication, under revision*) finding a value of 0.66. Such findings imply that the boulders analyzed in our image around the impact site are not different from the boulders located on the entire surface of Dimorphos, meaning that there should not be a sampling bias due to the boulder shape. In addition, as reported in Colas et al., paper (*Nature Communication, under revision*), such axial ratio value agrees with what found on other asteroids (Itokawa, Ryugu and Bennu) implying that Dimorphos boulders are similar in shape to the other NEAs' boulders.

Minor comments line by line:

Line 132: The formulation “nonlinear temperature gradient” might be ambiguous because temperature gradients can occur in different forms at airless planetary bodies, such as vertical thermal gradients, lateral thermal gradients, and thermal gradients in the first few millimetres of the uppermost regolith. Please specify the type of gradient.

The Referee is right, By “... a nonlinear temperature gradient ...”, we intended to imply all the various types of temperature gradients on the boulders, including the vertical and lateral thermal gradients. We have decided to rephrase it to “... a complex thermal insolation ...” to resolve the ambiguity.

Based on Editor’s suggestions, we expand the Methods section (especially the Thermophysical modelling of binary asteroids paragraph) retaining equations and details.

Line 160: Figure 1. I think it would be beneficial to indicate the locations of the boulders described in the text.

We reported a Figure in the Supplementary Information where the location of the boulders is indicated by ellipses. We replace Figure 1c with such figure for completeness, as reported below and modified the caption accordingly.

Line 178. Please also check if Boulder 1 and Boulder 2 carry a name. The USGS already lists several boulder names, such as Bodhran Saxum, Caccavella Saxum, and Puniu Saxum.
<https://astrogeology.usgs.gov/news/nomenclature/five-names-approved-for-dimorphos>

We thank the Referee for the suggestion, however such boulders have not been named yet.

Line 180: Please include a reference to the shape model.

We added the following citation: Daly et al., (2022), Ref. 25.

Line 187: The thermal inertia values of $370 \text{ Js}^{-1/2}\text{m}^{-2}\text{K}^{-1}$ and $1000 \text{ Js}^{-1/2}\text{m}^{-2}\text{K}^{-1}$ are reasonable.

We thank the Referee.

Line 216: Figure 3 provides the average temperatures of Atabaque Saxum. However, different orientations of the boulder's surface may lead to different local temperatures. I think it would be very instructive to plot the shape model of the boulder overlaid with a temperature map that highlights the complex spatial distributions of the temperature. The best would be a short sequence with three images: morning, midday, and afternoon.

We thank the Referee for the comment. We report below and added in main text the images requested showing the variation of Atabaque Saxum temperature map. Panels a), c), and e) show Atabaque's morning (8 AM), afternoon (4 PM), and midnight (12 AM) temperatures at periapsis, while panels b), d), and f) are the same but for apoapsis.

Line 246 - 248: What does "This" mean? Please specify: This argument..., this reasoning...

We specified, as suggested.

Line 250: I think it should be $t_{B,x} \sim 0.01-0.1t_{B,z}$ instead of $t_{B,x} \sim 0.01-0.1t_{B,x}$ because you compare the lateral and the vertical fracture propagation time.

The Referee is right, we corrected it in the text.

Line 262: Please explain Figure Caption 5 of what P is.

We specified it in the Figure's caption, P is the rotation period.

Line 254: "This timescale can be as low as 10 kyr..." I understand that Figure 6 a) presents the cumulative density function of the fracture length for $370 \text{ Js}^{-1/2}\text{m}^{-2}\text{K}^{-1}$, and the model values for 50 kyr and 100 kyr well align with the data. However, I struggle to follow your argumentation for Figure 6 b with $1000 \text{ Js}^{-1/2}\text{m}^{-2}\text{K}^{-1}$. Where does your conclusion "This timescale can be as low as 10 kyr" come from, and where do I find this in Figure 6 b? Please elaborate.

The Referee is right in raising this point because the above statement is not clear. The Figure 6 corresponds to the average of fractures propagation at aphelion and perihelion with a 3:1 weight for the two cases of thermal inertia. If we consider the perihelion case with $1000 \text{ Js}^{-1/2}\text{m}^{-2}\text{K}^{-1}$, as shown in the graph below, the horizontal fractures propagate and break the rocks in 10 kyr. We specify that the low timescale of 10 kyr corresponds to the perihelion case.

Line 263: Please explain what scaling means in this context.

We scaled the result obtained for ordinary chondrites obtained in Ref. 11 to the Didymos-Dimorphos conditions.

Line 280: "... because we neglect thermal stress between the elements of the rocks." Please mention this aspect earlier.

We report this aspect earlier at line 193.

Line 284-341: Please revise the discussion according to the new study design.

We revised the Methods section about the lineaments illumination bias and modify a few sentences in the main text. We changed and improved the Supplementary Figure showing the rendering of Bennu's boulder representing the different illumination condition and report the statistical analysis performed in the Supplementary Information (as stated above in the answer to the major points)

Line 414: Which kind of potential? Please specify.

We expand the Method section to clarify the statement.

Line 427: Please provide a numerical value that

We expand the Method section to clarify the statement.

Line 444: The authors mention that their thermal model uses direct and diffuse self-heating. Self-heating is a comparatively complex problem requiring more details: What is the bolometric albedo for self-heating? This quantity might be different from the bolometric albedo that determines the solar flux the surface absorbs. Self-heating is used to compute the temperature map T of the surface but already requires the temperature distribution. Consequently, it is a coupled problem. Do you use a full numeric solution to approach self-heating, or do they use a single-scattering approximation?

We appreciate the Reviewer's comments. We use the bolometric albedo of 0.067 in the thermophysical model and employ a ray-tracing technique to simulate scattering of solar radiation. We consider the single-scattering mode. In this mode, the diffuse self-heating contribution is assumed to be negligible. We have greatly expanded Methods section for the thermophysical model, which now provides this information.

Line 463: I think the full name is the Paris-Erdogan law.

We corrected it in the text.

Reviewer #3 (Remarks to the Author):

This manuscript presents a work regarding new findings on the asteroid Dimorphos that could have important implications on the activity and long term evolution of this type of asteroid, especially in a binary system from which no in-situ mission data existed. Although the thermal fracturing process underlined by authors has been already recently described on other near Earth asteroids, their findings apply on S-type asteroid for which no fractures observation existed before. Moreover, authors' study seems to confirm that this process is indeed pervasive and of universal relevance, at least for all rubble pile structured asteroids, and it is especially striking in the case of Dimorphos, being of a very small size. The manuscript is clear and to the point, with relevant figures supporting the claims. Interpretations are convincing and their implications are adequately discussed. However, considering the data shown and their limitations, I think that authors should moderately tone down their claims on the meridian direction of the observed fractures. Though, I consider that it won't change the overall meaning and relevance of this study. For the above reasons, I think this work will be worth to be published in Nature Communication and recommend these minor revisions to be addressed before final acceptance.

We thank the Referee for the positive revision and for the constructive comments and suggestions. As in line 134, we moderate the tones of the sentence regarding the directionality of fractures. If the Referee feels that the manuscript needs to tone down more our sentences, please tell us in order to change it accordingly. In the meantime, as reported below, we added in the Supplementary Information addition analyses that support the validity of the preferred orientation of Dimorphos fractures. Below, we addressed all the points raised by the Referee.

Detailed comments and minor comments:

- Throughout the manuscript the authors use the word “meridional” to qualify the “N-S/along meridian” directions of the thermal fractures. If I'm not mistaken the valid word should be “Meridian” as “meridional” meaning is “in the southern part”.

We thank the Referee for raising it, we checked and the term “meridional” can be used to indicate a pattern from north to south, or from south to north. We keep the “meridional” term in the text, if the Referee agrees.

- In Line 121, authors mention the fracture length to boulder size ratio, which is later used to show that all fractures don't cross through whole boulders. Unless I missed it, I could not find in the main text nor methods parts how this boulder size is defined? Is it the longest dimension of the boulder on the image? Or the major axis length of an ellipse fitted to each boulder? Authors should clarify this point.

Thank you.

The Referee is right, the size of the boulder is defined as the value of ellipse's major axis, that is used as the maximum diameter, as reported in Pajola et al., (*Nature Communication, under revision*). We clarified it in the caption of Table 1, which is reported in Supplementary Information.

- In Line 233 and in Figure 6 caption, unless I didn't understand well, authors mention “horizontal fractures”. As stated by authors in other part of the manuscript and as visible by interpreting the fractures trace onto the boulder geometry, most-to-all of the observed fractures seems to be vertical, but propagating preferentially in the horizontal direction.

Please clarify this point and modify these sentences accordingly.

From the mapping alone and due to the resolution of the image, it is not easy to determine if fractures are propagating vertical or not, but the modelling results suggest that the mapped fractures are horizontally propagating. In that sense, we refer to horizontal fractures to those

Editorial Note: Figure below from Delbo, M., Walsh, K.J., Matonti, C. et al. Alignment of fractures on Bennu's boulders indicative of rapid asteroid surface evolution. *Nat. Geosci.* **15**, 453–457 (2022), <https://doi.org/10.1038/s41561-022-00940-3>, reproduced with permission from Springer Nature

fractures that propagate horizontally, as shown in the Figure below (Ref. 11). From our thermal fatigue modelling, we found that most of mapped fractures on Dimorphos' boulders should be shallow and propagate along the boulder surface on Dimorphos (not in the vertical direction), a process that may occur one to two orders of magnitude faster than in vertical direction.

- At line 146 authors validly mention that thermal polygonal fractures on 67P have been shown to follow “exponential size distribution” and quote the reference 32 following this claim. To my knowledge, this article does not mention the size distribution of 67P polygonal fractures. However, such exponential distribution of polygonal fractures on 67P has been underlined in the following work. <https://doi.org/10.1038/s41561-019-0307-9>.

The Referee is right, we changed the citation accordingly.

- Throughout the manuscript, the authors state that their analysis clearly shows that fractures are oriented following a “meridian” direction. I think their interpretations is likely valid and well discussed in the method and Sup Material parts. However, considering the small number of interpreted fractures and the possible observation bias that authors validly explain for E-W fracture, I feel that authors need to be a bit more cautious and could use more conditional in the main text too. For instance, by stating that fractures directions are surely “Non-random”, which won't change the meaning and relevance of the results. Indeed, I think the non-randomness of fracture direction is the most important point here. Moreover, it could be interesting to discuss the fact that E-W trending fractures could indeed exist, without necessarily impeding the thermal origin of the observed fractures, since other directions could result from previous shifts in the spin axis of Dimorphos. Speaking of which, it may be interesting to discuss possible shift of the spin axis in the past? And is it expected to be rather unstable, considering the binary nature of this system?

We thank the Referee for the comment. We added in the Supplementary Information the analyses we have accomplished to strengthen the validity of our findings (preferred orientation of boulder's fractures). In particular, we investigate:

(i) the probability that the observed azimuthal distribution arose by chance from a uniform Distribution, i.e. from randomly oriented fractures;

(ii) the sampling bias effect, implying that fractures with azimuth similar to the sunlight direction are less likely to be identified, and thus are not present from the dataset.

The simulation reported in the Supplementary Information supports the following statements:
(i) the dataset is unlikely to be simply the result of a uniform distribution, hence leading to the rejection of the null hypothesis (the dataset comes from a uniform distribution).

(ii) the dataset is unlikely to result from a uniform distribution altered by the illumination bias alone.

We highlight that from a statistically perspective, such a result does not prove that the dataset cannot be derived from the superimposition of a uniform distribution process and additional

effect, rather than our original dataset does not directly come from a uniform distribution, neither from a uniform distribution plus the illumination bias.

Regarding the stability of the Didymos system, it is assumed that the mutual orbit and the secondary spin were not excited upon arrival at the binary system. Indeed, the system's pre-impact dynamical state was assumed relaxed, meaning the mutual orbit was well-circularized with the secondary in the 1:1 spin orbit resonance and any free libration was minimized (Agrusa et al., 2021, 2024, Richardson et al., 2022).

Reviewer #3 (Remarks to the Author):

Dear Authors, see my comments in the attached file. Thank you.

Reviewer #3 (Attachment):

After this first round of revision, I am still favourable to the publication of this manuscript in Nature communication. Though, I recommend these last modest points to be addressed by the authors before publication.

I list below (in blue) my comments addressing some of the authors answers:

- We thank the Referee for raising it, we checked and the term “meridional” can be used to indicate a pattern from north to south, or from south to north. We keep the “meridional” term in the text, if the Referee agrees.

It is indeed, but I think that the double meaning of “meridional” could be misleading for the reader, especially as, to my knowledge, meridional is less used than “meridian” to mean “along meridian/North-south direction”. Though I won’t argue much about it and let the editor/authors decide on this minor point.

- The Referee is right, the size of the boulder is defined as the value of ellipse’s major axis, that is used as the maximum diameter, as reported in Pajola et al., (Nature Communication, under revision). We clarified it in the caption of Table 1, which is reported in Supplementary Information.

All good, thank you.

- From the mapping alone and due to the resolution of the image, it is not easy to determine if fractures are propagating vertical or not, but the modelling results suggest that the mapped fractures are horizontally propagating. In that sense, we refer to horizontal fractures to those fractures that propagate horizontally, as shown in the Figure below (Ref. 11). From our thermal fatigue modelling, we found that most of mapped fractures on Dimorphos’ boulders should be shallow and propagate along the boulder surface on Dimorphos (not in the vertical direction), a process that may occur one to two orders of magnitude faster than in vertical direction.

(I apologize in advance if I didn’t understand the authors answer)

I understand, but as authors show in the figure from Delbo et al., the fracture shown here is vertical. From a structural geology point of view or other material-science domains which study fractures, fractures planes are defined from their dip angle (angle to the g vector or local surface) and strike (angle to the North or other local reference direction). In your example, although they propagate preferentially in the horizontal direction, fracture “dip angle” is sub vertical, therefore these fractures are vertical or sub-vertical, and cannot be described as horizontal, although they propagate more in the horizontal direction and are shallow. I think the problem here comes from confusing direction of propagation with the angle of the fracture plane.

Finally, (at least for the longest ones), in my opinion fractures dip are quite clearly subvertical when we observe the way they cross between neighbour boulders’ faces. Consequently, I think authors should, if not calling them vertical, at least remove the “horizontal” word in qualifying these fractures, as it is confusing. Or they could use “horizontally propagating fractures”.

- We thank the Referee for the comment. We added in the Supplementary Information the analyses we have accomplished to strengthen the validity of our findings (preferred orientation of boulder's fractures). In particular, we investigate:

(i) the probability that the observed azimuthal distribution arose by chance from a uniform Distribution, i.e. from randomly oriented fractures;

[...]

We thank the Referee for the positive revision and for the constructive comments and suggestions. As in line 134, we moderate the tones of the sentence regarding the directionality of fractures. If the Referee feels that the manuscript needs to tone down more our sentences, please tell us in order to change it accordingly. In the meantime, as reported below, we added in the Supplementary Information addition analyses that support the validity of the preferred orientation of Dimorphos fractures. Below, we addressed all the points raised by the Referee.

I see, thank you for this newly added and thorough analyses, it clarifies your interpretation. However, in my opinion, the changes done in the main text are a bit too small. Indeed, authors shows that the distribution of fractures directions is very likely a "non-random" distribution. Considering the statistically small number of interpreted fracture, along with the restricted zone on dimorphos where these interpretations have been possible I think it would be fair to clearly state that in the main text (at least the first time this fact is mentioned), i.e. that these fractures are shown to have a "non-random" orientation distribution, and are possibly oriented preferentially along a North/South direction. Which in my opinion is enough to prove that these fractures don't originate from impact or boulders tumbling/movement, thus are from a thermal origin.

After this first round of revision, I am still favourable to the publication of this manuscript in Nature communication. Though, I recommend these last modest points to be addressed by the authors before publication.

We thank the referee for the positive and constructive feedback. We report below in bold our responses to referee's requests.

I list below (in blue) my comments addressing some of the authors answers:

- We thank the Referee for raising it, we checked and the term "meridional" can be used to indicate a pattern from north to south, or from south to north. We keep the "meridional" term in the text, if the Referee agrees.

It is indeed, but I think that the double meaning of "meridional" could be misleading for the reader, especially as, to my knowledge, meridional is less used than "meridian" to mean "along meridian/North-south direction". Though I won't argue much about it and let the editor/authors decide on this minor point.

We thank the referee, but we prefer to maintain the "meridional" term.

- The Referee is right, the size of the boulder is defined as the value of ellipse's major axis, that is used as the maximum diameter, as reported in Pajola et al., (Nature Communication, under revision). We clarified it in the caption of Table 1, which is reported in Supplementary Information.

All good, thank you.

- From the mapping alone and due to the resolution of the image, it is not easy to determine if fractures are propagating vertical or not, but the modelling results suggest that the mapped fractures are horizontally propagating. In that sense, we refer to horizontal fractures to those fractures that propagate horizontally, as shown in the Figure below (Ref. 11). From our thermal fatigue modelling, we found that most of mapped fractures on Dimorphos' boulders should be shallow and propagate along the boulder surface on Dimorphos (not in the vertical direction), a process that may occur one to two orders of magnitude faster than in vertical direction.

(I apologize in advance if I didn't understand the authors answer)

I understand, but as authors show in the figure from Delbo et al., the fracture shown here is vertical. From a structural geology point of view or other material-science domains which study fractures, fractures planes are defined from their dip angle (angle to the g vector or local surface) and strike (angle to the North or other local reference direction). In your example, although they propagate preferentially in the horizontal direction, fracture "dip angle" is sub vertical, therefore these fractures are vertical or sub-vertical, and cannot be described as horizontal, although they propagate more in the horizontal direction and are shallow. I think the problem here comes from confusing direction of propagation with the angle of the fracture plane.

Finally, (at least for the longest ones), in my opinion fractures dip are quite clearly subvertical when we observe the way they cross between neighbour boulders' faces. Consequently, I think authors should, if not calling them vertical, at least remove the "horizontal" word in qualifying these fractures, as it is confusing. Or they could use "horizontally propagating fractures".

We thank the referee for the comment, we totally agree with the structural geology point of view. Hence, to avoid misunderstanding we follow the referee's suggestions and change the term "horizontal" in "horizontally propagating fractures".

- We thank the Referee for the comment. We added in the Supplementary Information the analyses we have accomplished to strengthen the validity of our findings (preferred orientation of boulder's fractures). In particular, we investigate:

(i) the probability that the observed azimuthal distribution arose by chance from a uniform Distribution, i.e. from randomly oriented fractures;

[...]

We thank the Referee for the positive revision and for the constructive comments and suggestions. As in line 134, we moderate the tones of the sentence regarding the directionality of fractures. If the Referee feels that the manuscript needs to tone down more our sentences, please tell us in order to change it accordingly. In the meantime, as reported below, we added in the Supplementary Information addition analyses that support the validity of the preferred orientation of Dimorphos fractures. Below, we addressed all the points raised by the Referee.

I see, thank you for this newly added and thorough analyses, it clarifies your interpretation. However, in my opinion, the changes done in the main text are a bit too small. Indeed, authors shows that the distribution of fractures directions is very likely a "non-random" distribution. Considering the statistically small number of interpreted fracture, along with the restricted zone on dimorphos where these interpretations have been possible I think it would be fair to clearly state that in the main text (at least the first time this fact is mentioned), i.e. that these fractures are shown to have a "non-random" orientation distribution, and are possibly oriented preferentially along a North/South direction. Which in my opinion is enough to prove that these fractures don't originate from impact or boulders tumbling/movement, thus are from a thermal origin.

We changed such statement in section "Mapping and Analysis of boulders' fractures" as suggested by the referee (line 130-132).